# In vivo nuclear capture and molecular profiling identifies *Gmeb1* as a transcriptional regulator essential for dopamine neuron function

Luis M. Tuesta[1,2,3], Mohamed N. Djekidel[1,2,3,7], Renchao Chen[1,2,3,7], Falong Lu [1,2,3], Wengang Wang[1,4], Bernardo L. Sabatini [1,4] & Yi Zhang [1,2,3,5,6]

Midbrain dopamine (mDA) neurons play a central role in reward signaling and are widely implicated in psychiatric and neurodegenerative disorders. To understand how mDA neurons perform these functions, it is important to understand how mDA-specific genes are regulated. However, cellular heterogeneity in the mammalian brain presents a major challenge to obtaining this understanding. To this end, we developed a virus-based approach to label and capture mDA nuclei for transcriptome (RNA-Seq), and low-input chromatin accessibility (liDNase-Seq) profiling, followed by predictive modeling to identify putative transcriptional regulators of mDA neurons. Using this method, we identified *Gmeb1*, a transcription factor predicted to regulate expression of *Th* and *Dat*, genes critical for dopamine synthesis and reuptake, respectively. *Gmeb1* knockdown in mDA neurons resulted in downregulation of *Th* and *Dat*, as well as in severe motor deficits. This study thus identifies *Gmeb1* as a master regulator of mDA gene expression and function, and provides a general method for identifying cell type-specific transcriptional regulators.

[1] Howard Hughes Medical Institute, Boston, MA 02115, USA. [2] Program in Cellular and Molecular Medicine, Boston Children's Hospital, Boston, MA 02115, USA. [3] Department of Genetics, Harvard Medical School, Boston, MA 02115, USA. [4] Department of Neurobiology, Harvard Medical School, Boston, MA 02115, USA. [5] Division of Hematology/Oncology, Department of Pediatrics, Boston Children's Hospital, Boston, MA 02115, USA. [6] Harvard Stem Cell Institute, WAB-149G, 200 Longwood Avenue, Boston, MA 02115, USA. [7] These authors contributed equally: Mohamed N. Djekidel, Renchao Chen. Correspondence and requests for materials should be addressed to Y.Z. (email: yzhang@genetics.med.harvard.edu)

Midbrain dopamine (mDA) neurons account for less than 1% of all neurons in the brain[1]. Despite their limited number, mDA neurons are critical regulators of reward, cognition, and movement, and play a fundamental role in the pathophysiology of psychiatric and neurodegenerative disorders[2,3]. In contrast to the well-defined role of dopamine signaling in health and disease, relatively little is known about how chromatin architecture regulates mDA gene expression. This knowledge gap is largely due to the cellular heterogeneity in the brain which impedes isolation of pure mDA nuclei from neighboring cell types, and therefore, generation of an mDA-specific map of accessible chromatin from which to identify transcriptional regulatory elements.

Chromatin accessibility is positively correlated to transcriptional activity. Actively transcribed genes often possess accessible (open) chromatin regions at their promoters and enhancers, thus permitting the recruitment of transcriptional factors and co-activators, whereas genes associated with inaccessible (closed) chromatin are generally transcriptionally repressed[4]. As such, chromatin accessibility profiling has been used to gain insights into transcriptional regulation of various abundant cortical cell types during brain development[5,6].

In this study, we aim to understand how gene expression is regulated in mDA neurons, a limited, but important cell type. To this end, we developed a simple virus-based system to purify mDA nuclei from adult mouse brain for transcriptome and low-input chromatin accessibility profiling (Fig. 1a), and posit a predictive model of transcriptional regulation. As a proof of principle, we focus on one of the candidate transcription factors, Gmeb1. We provide evidence demonstrating that Gmeb1 plays an important role in regulating the expression of Th and Dat, genes critical for dopamine neuron function, and show that Gmeb1 is necessary for maintenance of motor coordination.

## Results

### Cre-inducible HA nuclear tagging facilitates purification of mDA nuclei.
To achieve mDA neuron-specific labeling, we designed a Cre-inducible AAV vector encoding the nuclear envelope protein KASH[7,8] with an HA tag (Fig. 1b, top panel). To test whether the infection is specific to mDA neurons, we injected the AAV-DIO-KASH-HA (KASH-HA) virus into the midbrain of dopamine transporter (Dat)-Cre heterozygous (het) mice. Dat-Cre mice have been routinely used for targeted gene expression in mDA neurons as Dat (also referred to as Slc6a3) is specifically expressed in mDA neurons[9]. Consequently, we expected that KASH-HA expression would be restricted to Cre-expressing mDA neurons. Immunostaining confirmed that KASH-HA injected Dat-Cre heterozygous mice exhibited robust expression of HA while no HA signal was detectable when the same virus was injected to wildtype (WT) mice (Fig. 1b, bottom panels). Quantification of co-staining of HA and tyrosine hydroxylase (Th), a rate-limiting enzyme in catecholamine synthesis which serves as an mDA marker, revealed that 87.2% of mDA neurons were HA positive (HA+), while 97.3% of HA+ cells were dopaminergic (Fig. 1c). Collectively, these data suggest that the expression of KASH-HA is efficient and is largely restricted to mDA neurons in Dat-Cre mice.

To test the feasibility of using this nuclear tagging technique to capture mDA nuclei for molecular profiling, we injected a separate cohort of Dat-Cre mice with KASH-HA virus and collected midbrain samples 14 days later. Immediately after collection, samples were processed for nuclear isolation, HA immunostaining, and fluorescence-activated nuclear sorting (FANS) to separate HA+ and HA− nuclei (Supplementary Fig. 1a). To confirm that HA+ nuclei represented mDA neurons, we performed RNA−seq on both the HA+ and HA− nuclei. Transcriptome analysis revealed that of the 13,727 genes expressed in HA+ and HA− nuclei (FPKM > 1), 394 genes exhibited at least a 4-fold enrichment in the HA+ cells, including known dopaminergic marker genes such as Th, Dat, Ddc, and Vmat[10]. Conversely, 953 genes were significantly depleted (<4-fold) in HA+ relative to HA− nuclei—including glial markers Gjb8 and Cldn11[11] (Fig. 1d). Gene ontology (GO) analysis (cutoff: $q < 0.05$) revealed that HA+ nuclei were enriched for mDA-related functions such as response to cocaine and DA neuron differentiation, while HA− nuclei were enriched in non-neuronal functions such as gliogenesis, glial cell differentiation, and axon ensheathment (myelination), further confirming the purity of the sorted HA+ nuclei (Fig. 1e, Supplementary Fig. 1b).

### Multi-omics analysis predicts Gmeb1 as an mDA transcriptional regulator.
The mDA transcriptome presented in Fig. 1d, while depleted of glial gene expression, still included genes commonly found in other neuron types. To identify genes highly enriched in mDA neurons, we first derived a consensus mDA transcriptome from RNA-Seq of two biological replicates of purified mDA nuclei (Supplementary Fig. 2a) and then compared their expression level against three cortical neuron subtypes: vasoactive intestinal protein (VIP)-cortical, excitatory-cortical (Exc)-cortical, and parvalbumin (PV)-cortical neurons[5]. This analysis revealed that out of the 394 HA+ genes, 107 are mDA-enriched (at least 4-fold higher in mDA neurons compared with VIP, Exc, and PV neurons, and $q$-value < 0.001) (Fig. 2a, b, Supplementary Fig. 2b, Supplementary Data 1). As expected, the list included mDA signature genes Th, Dat, Ddc, and Vmat. However, some of the HA+ enriched genes identified in Fig. 1d, such as Nurr1 and Satb1, were also highly expressed in cortical neuron subtypes and were thus excluded from the list of mDA-enriched genes (Supplementary Data 2). GO term and pathway analysis of the 107 genes further confirmed their enrichment in mDA neuron-specific functions such as dopamine metabolism, transport, and secretion (Supplementary Fig. 2c). Thus, we defined these 107 genes as "mDA-enriched genes".

Next, we attempted to identify transcriptional regulators involved in the activation of these 107 mDA-enriched genes. To this end, we performed liDNase-Seq[12], a technique that allows genome-wide identification of transcriptional regulatory elements using limited cell numbers. Using two biological replicates of 500 mDA nuclei (Supplementary Fig. 3a), we mapped the mDA DNaseI-hypersensitive site (DHS) landscape. Among the 28,084 detected DHSs ($p < 0.05$), a large portion (over 40%) of these accessible chromatin sites were located in gene promoter regions (within ± 3 kb from transcription start site-TSS) (Supplementary Fig. 3b), consistent with previous reports in other cell types[13,14]. By comparing mDA neuron DHS sites with the chromatin accessible sites of cortical neurons[5], we identified 2374 "mDA-enriched DHSs": open chromatin sites present in mDA neurons but absent from cortical neurons (Fig. 2c). Interestingly, the majority of the "mDA-enriched DHSs" were localized in non-promoter regions (3 kb away from TSS) (Supplementary Fig. 3c), highlighting the importance of distal DHSs in defining cell identity, in agreement with previous observations[6,15]. Consistent with the positive correlation between chromatin accesibility and transcription, genes containing promoter DHSs in mDA neurons showed significantly higher expression levels compared with the genes lacking promoter DHSs ($p$-value = 2.2e−16, two-tailed Mann–Whitney–Wilcoxon Test), and this correlation was also maintained in mDA-enriched genes ($p$-value = 0.005397, two-tailed Mann–Whitney–Wilcoxon Test) (Supplementary Fig. 3d).

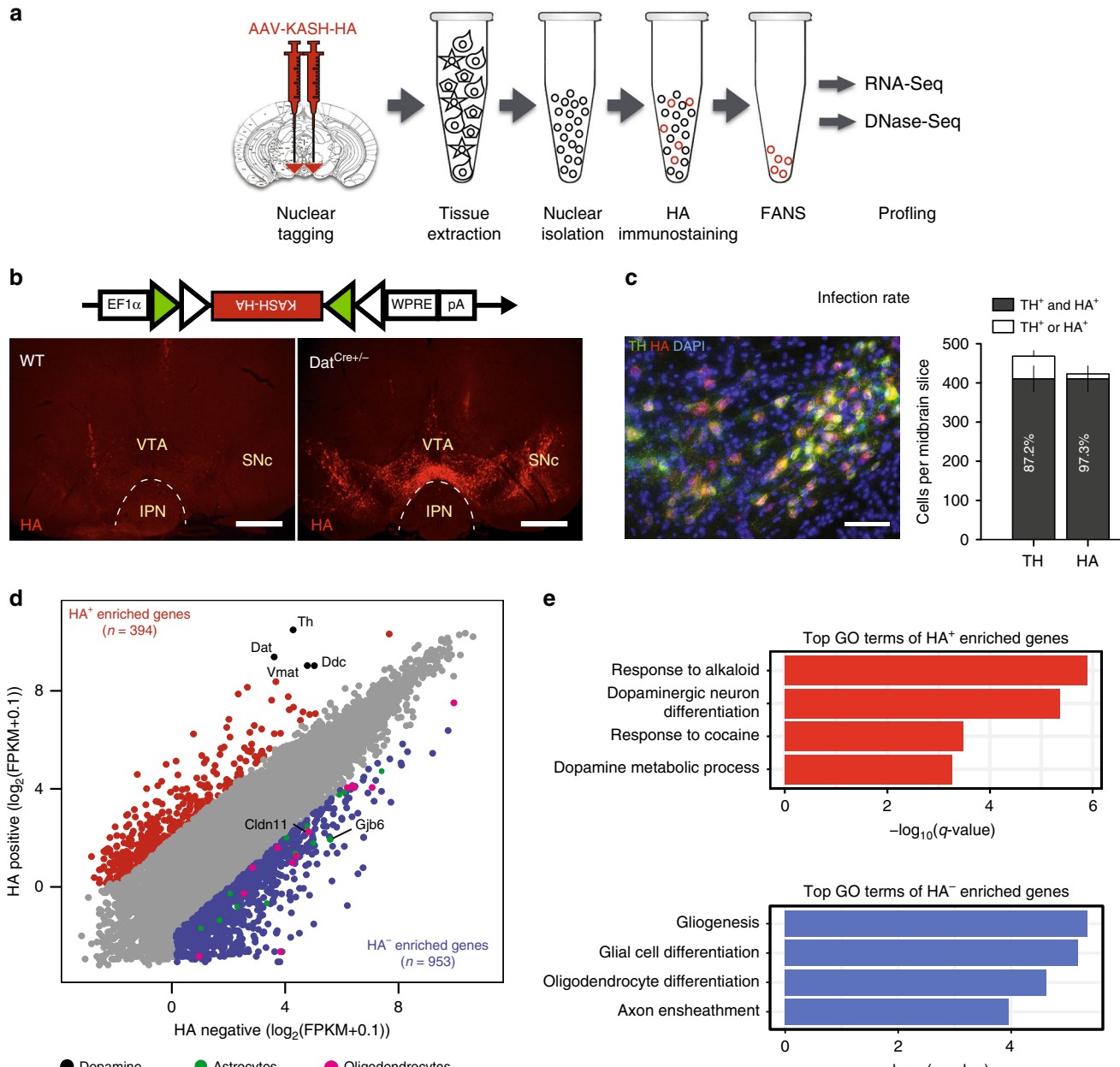

**Fig. 1** Neuron subtype-specific nuclear capture and transcriptome analysis. **a** Schematic illustration of the experimental steps for mDA nuclear capture. AAV-DIO-KASH-HA (DIO-KASH-HA) is bilaterally injected into the midbrain of Dat-Cre het mice. Two weeks later, midbrains are dissected and nuclei are isolated by Fluorescence Assisted Nuclear Sorting (FANS). The isolated nuclei were then used for multi-omics analysis. Note: Needle/syringe image was adapted from Keynote clipart, and coronal brain section was reproduced from ref. [48]. (Copyright 2013, Elsevier, Academic Press). **b** Diagram of the DIO-KASH-HA vector (top) and representative micrographs illustrating infection specificity to Cre-expressing neurons. WT mice injected with DIO-KASH-HA virus showed no HA signal (bottom left) as opposed to Dat-Cre het mice (bottom right). Scale bar: 500 μm. **c** Merged representative micrograph (left) showing near-complete colocalization of HA signal (red) with TH+ neurons (green). Scale bar: 50 μm. Histogram (right) illustrating infection efficiency (mean ± s.e.m.) of DIO-KASH-HA construct ($n = 6$). 87.2% of DA neurons were infected by DIO-KASH-HA, while 97.2% of DIO-KASH-HA-infected neurons were DAergic. **d** Scatter plot comparing RNA-seq data of HA+ and HA− nuclei indicating HA+ nuclei are enriched for DAergic markers (black), and depleted of astrocyte (green) and oligodendrocyte (magenta) markers. HA+ nuclei enriched genes are marked in red. Segregation criteria: fold-change > 4; FPKM > 1. **e** Enriched GO terms associated with the 394 HA+ and 953 HA− nuclei enriched genes, respectively ($q$-value < 0.05)

Out of the 107 mDA-enriched genes, 59 contained promoter DHSs. To identify candidate transcription factors (TFs) that could regulate these mDA-enriched genes, we performed genomic sequence motif enrichment analysis at promoter DHSs and identified 11 TF-binding motifs ($p$-value < 0.0001, FPKM > 1) (Fig. 2d, Supplementary Fig. 3e). This analysis not only identified known regulators of mDA gene expression such as *Clock*[16] and

*Creb1*[17], but also *Gmeb1* (glucocorticoid modulatory element binding protein-1). To ensure that *Gmeb1* was not an artifact of transcriptome pruning, we performed a similar analysis using the 394 HA+ enriched gene promoters, and it also identified *Gmeb1* (Supplementary Fig. 4b). Interestingly, *Gmeb1* is a TF not previously known to play a role in mDA gene expression, whose binding motif is not present among cortical neuron-enriched

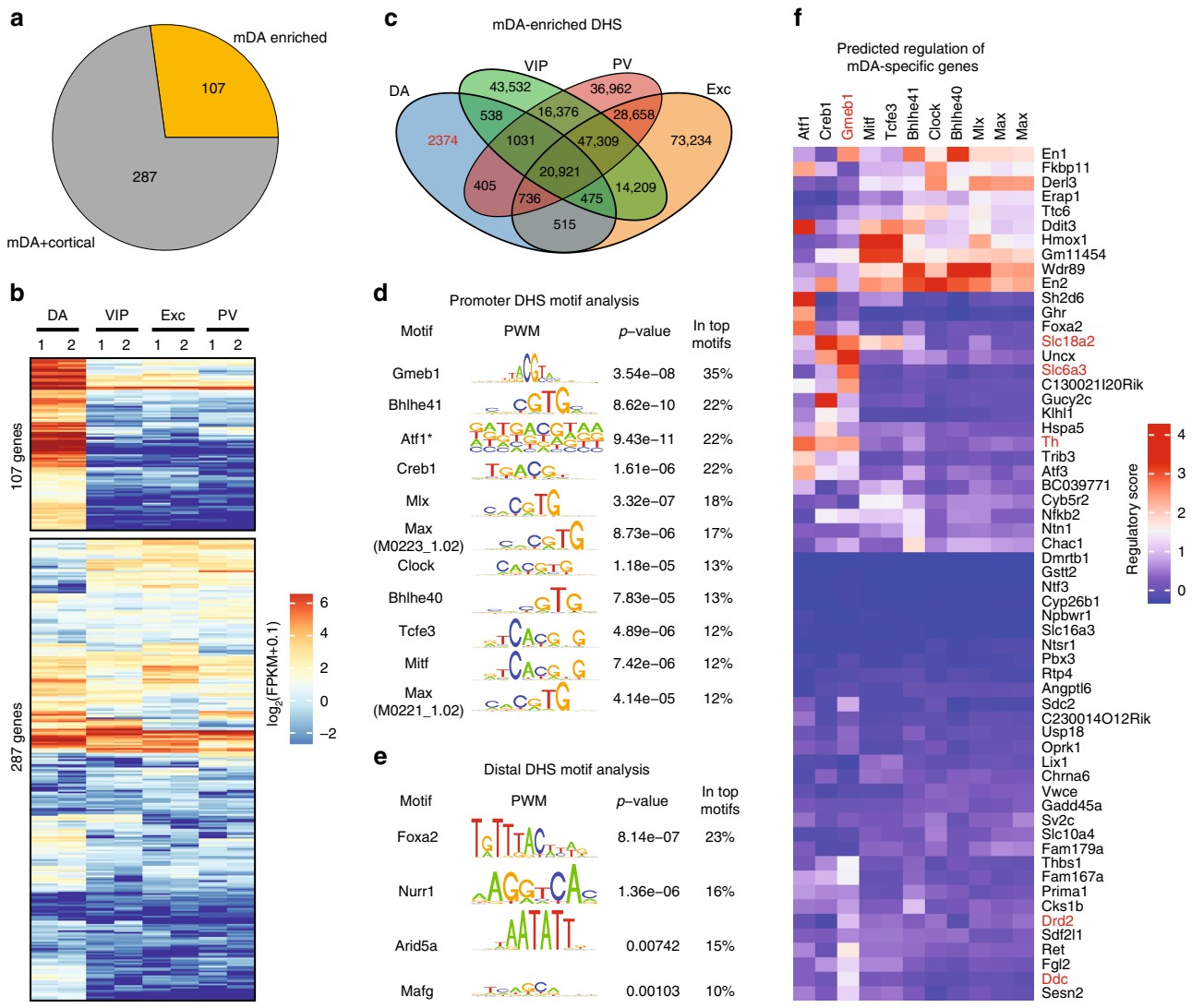

**Fig. 2** Transcriptome and DHS analysis reveals putative mDA transcriptional regulators. **a** Pie chart showing the number of genes significantly enriched in HA⁺ mDA neurons compared with VIP, Exc, and PV cortical neuron-expressed genes (FC > 4 and q-value < 0.001, FPKM > 1). **b** Heatmap showing the 394 genes, respectively, expressed in the HA⁺ mDA neurons, as well as VIP, Exc, and PV cortical neurons. The 107 genes significantly enriched in mDA neurons (top) are termed "mDA-enriched" genes. **c** Venn diagram showing the overlap between the 28,084 mDA neuron DHS sites (q-value < 0.05) with the ATAC-Seq accessible chromatin regions (q-value < 0.05) in VIP, Exc and PV cortical neurons. The 2374 DHS sites present only in mDA neurons were termed "mDA-enriched DHS". **d** List of the motifs identified at the promoter DHS of mDA-enriched genes (q-value < 1e−4). Each base-pair of the motif is proportional to its information content profile[49]. The Atf1 motifs with a * to indicate the low information content along all the positions, thus its motif was instead represented by the relative base frequency. The identifier under the Max motif indicate the motif PWM ID in the CIS-BP database[50]. **e** List of motifs identified at mDA-enriched distal DHSs of mDA-enriched genes (q-value < 0.01). Each base-pair of the motif is proportional to its information content profile[49]. **f** Heatmap showing the predicted TF regulatory score (TFRS) associating each TF to the mDA-enriched genes with a promoter DHS

promoters (Supplementary Fig. 4a, c–e, Supplementary Data 3) despite its expression in cortical neurons[5]. Surprisingly, *Nurr1* and *Foxa2*, two TFs extensively studied for their role in mDA neuron development and function[18,19], were absent from mDA promoter motif analysis. However, these TFs were identified as the top two candidates when a motif analysis was performed on distal DHSs (3 kb–1 Mb from TSS) of mDA-enriched genes (Fig. 2e, Supplementary Fig. 5, Supplementary Data 4).

To better understand the potential role of each of the 11 TF candidates in regulating the 59 mDA-enriched genes with accessible promoters, we assigned a "regulatory score" to each TF-gene pair where a higher score reflects a greater likelihood that the given TF would have the potential to regulate a given target gene through promoter DHS binding (Fig. 2f). To generate this

predictive model, we designed four regulatory scoring schemes (Eqs. 1–4 in Methods) that associate gene expression with different TF-binding features to predict the regulatory effect for each of the 11 TFs on the expression level of the 59 genes. For each model we calculated the prediction error relative to the true expression level of the 59 genes (data used in Fig. 2b). We found that the most predictive scheme is the one that considered the TF binding affinity to the promoter DHS (represented by the binding *p-value*) and the distance of the DHS to the TSS (Supplementary Fig. 6, equation (3), see methods for details). Using this scheme, *Gmeb1* was predicted to regulate mDA "identity" genes, including those involved in dopamine synthesis (*Th*, *Ddc*), dopamine vesicular packaging (*Slc18a2*), dopamine reuptake (*Slc6a3*), and autoregulation (*Drd2*) (Fig. 2f, Supplementary Data 5).

**Gmeb1 regulates transcription of mDA identity genes**. *Gmeb1* has been shown to increase sensitivity to low glucocorticoid concentrations by acting as a transcription factor at the tyrosine transaminase promoter[20], and has also shown to be a neuroprotective factor against oxidative stress[21], but its role in mDA neuron function has not been previously implicated. Considering that the *Gmeb1* binding motif is present in 35% of the accessible promoters of mDA-enriched-specific genes (Fig. 2d) and that two of the key mDA genes, *Th* and *Dat*, are predicted to be its targets, we hypothesized that *Gmeb1* plays an important role in mDA neuron function. To test the transcriptional effects of *Gmeb1* knockdown in mDA neurons, we designed three shRNAs targeting *Gmeb1* and assessed their knockdown efficiency in N2A cells (Supplementary Fig. 7a, b). We then packaged the most efficient shRNA (sh1) into a vector (AAV-DIO-KASH-GFP-U6-shRNA), and delivered it into the midbrain of Dat-Cre mice (Fig. 3a). Two weeks after injection, midbrain tissue was dissected, and nuclei were isolated and immunostained for GFP. GFP+ (mDA) nuclei were FANS sorted and used for RNA-Seq to assess the transcriptional effects of Gmeb1 depletion. Transcriptome analysis of two biological replicates of control (shScramble) and *Gmeb1* knockdown samples demonstrated high reproducibility (Supplementary Fig. 7c). *Gmeb1* knockdown resulted in downregulation and upregulation (FC > 2) of 99 and 78 genes, respectively, in mDA neurons (Fig. 3b, Supplementary Data 6 and 7). The down-regulated genes included 9 mDA-enriched genes (*Dat, Th, Cnpy1, Avpr1a, Gucy2c, Aldh1a1, Ndnf, Anxa1, Chrna6*). Notably, both *Th* and *Dat*, whose promoters contain Gmeb1 binding motifs, were significantly down-regulated following *Gmeb1* knockdown (Fig. 3c). Immunostaining further confirmed depletion of *Th* and *Dat* at the protein level, in both midbrain and dorsal striatum, two principal projection regions for mDA neurons (Fig. 3d, e, Supplementary Fig. 8). Interestingly, a weak KASH signal was also detected in Th⁻ neurons in the shScramble group, suggesting that the vector may also be expressed in some non-mDA cells (Fig. 3d). This could be attributed to AAV design, where the knockdown vector (shScramble and shGmeb1) (Fig. 3a) contained a different backbone and promoter than the one used for nuclear tagging (Fig. 1b).

To ensure that the lack of Th or Dat protein signal was not due to cell death resulting from *Gmeb1* knockdown, we compared the transcriptome of mDA neurons 2 weeks after infection with viruses expressing shGmeb1 or shScramble. We found that none of the 7 queried apoptosis-related genes showed significant alteration in *Gmeb1* knockdown mDA neurons compared with the shScramble controls (Supplementary Fig. 9a). This result is consistent with the lack of significant increase in the number of cleaved caspase-3 positive mDA neurons (Supplementary Fig. 9b, c). However, these results cannot rule out the possibility that inflammatory processes may induce cell death by necrosis, or that cells may have already degenerated through apoptosis by this timepoint, leaving only the surviving cells for transcriptome analysis.

To address these possibilities, we first assessed the expression levels of genes involved in necrosis and found no significant alteration in response to *Gmeb1* knockdown (Supplementary Fig. 9d). Next, we co-injected viruses expressing either shGmeb1 and cre-inducible mCherry (DIO-mCherry), or shScramble and DIO-mCherry into the substantia nigra pars compacta (SNc) of Dat-Cre mice. This approach ensured that mDA neurons could be identified and counted despite Th signal loss following *Gmeb1* knockdown. We found that *Gmeb1* knockdown in the SNc, while resulting in Th loss (Supplementary Fig. 10a), does not change the number of mDA neurons (mCherry⁺), thus suggesting no mDA neuron loss (Supplementary Fig. 10c). This inference was

further supported by TUNEL assay, in which fragmented DNA, a hallmark of cell death, was rarely detected in either shGmeb1 or shScramble group (Supplementary Fig. 10a). In contrast, treatment with DNase-1, which induces DNA fragmentation, resulted in robust TUNEL signal (Supplementary Fig. 10a), suggesting that loss of *Gmeb1*, while abrogating *Th* and *Dat* expression, does not induce cell death.

While *Gmeb1* knockdown does not result in mDA neuron death, the transcriptional consequence of this manipulation could alter the basic electrophysiological properties of mDA neurons due to its effect on dopaminergic transmission. To explore this possibility, we tested the effect of *Gmeb1* knockdown on the excitability of SNc mDA neurons by current clamp, using the viral injection strategy described in Supplementary Fig. 10b. We found that loss of Gmeb1 did not significantly change the nature or frequency (Hz) of evoked action potentials (Fig. 4a, Supplementary Fig. 11) when compared with shScramble controls, nor the amplitude (mV) of the recorded potentials (Fig. 4b, Supplementary Fig. 11a). Furthermore, hyperpoloarization currents resulted in similar sag potentials in both groups (Supplementary Fig. 11b). Collectively, these results suggest that loss of Gmeb1 does not affect SNc mDA neuron excitability, firing rate or magnitude of evoked action potentials.

**Gmeb1 is required for maintaining homeostatic motor coordination**. Tyrosine hydroxylase (Th) is the rate-limiting enzyme in catecholamine synthesis[22] and therefore depletion of Th results in loss of dopamine[23]. Dysregulation of dopamine signaling in humans has been associated with mood disorders, drug addiction and is the root cause of motor impairments associated with Parkinson's disease (PD)—a neurodegenerative disorder characterized by progressive loss of nigral mDA neurons, resulting in reduced motor coordination, balance and increased muscle fatigue[2,3]. Given that *Gmeb1* plays a critical role in regulating *Th* expression, we next asked whether *Gmeb1* would be necessary to maintain normal motor functions. To this end, we tested mice with bilateral SNc *Gmeb1* knockdown in a battery of motor assessments, including the pole test, rotarod test, swim test and hanging wire test. In the pole test, a mouse is required to walk down a 50-cm grooved vertical pole from top to base, and animals with balance impairments will take longer to reach the base of the pole. We found that SNc *Gmeb1*-knockdown mice (shGmeb1) took more time to reach the base than control (shScramble) mice (Fig. 5a, Supplementary Movie 1; $n = 11$/group, ****$p < 0.0001$ with two-tailed *t*-test). Consistent with pharmacological models of dopamine depletion[24,25], our results suggest that *Gmeb1* knockdown in SNc results in balance and coordination impairments.

The rotarod test, which requires mice to continuously walk on an accelerating rod over the span of 5 min is also used to measure balance and coordination[26], and the "latency"—the amount of time the mouse stays on the rod without falling or clinging on, is recorded. We found that *Gmeb1* knockdown mice exhibited lower latency than control mice (Fig. 5b, Supplementary Movie 2, $n = 11$/group, ****$p < 0.0001$ with two-tailed *t*-test). This result indicates that *Gmeb1* knockdown impairs rotarod performance at lower speeds, and thus suggests a deleterious effect on balance and coordination, consistent with results of the pole test (Fig. 5a). However, since climbing and trotting are behaviors in which nearly all experimental mice have daily experience, we tested the animals' ability to adapt to a new environment by performing an innate behavior in which they had no prior experience. To this end, we placed mice in a water-filled chamber and assessed the animals' swimming ability over the span of 5 min. Using a scale of 0–5 where the higher score indicates greater swimming

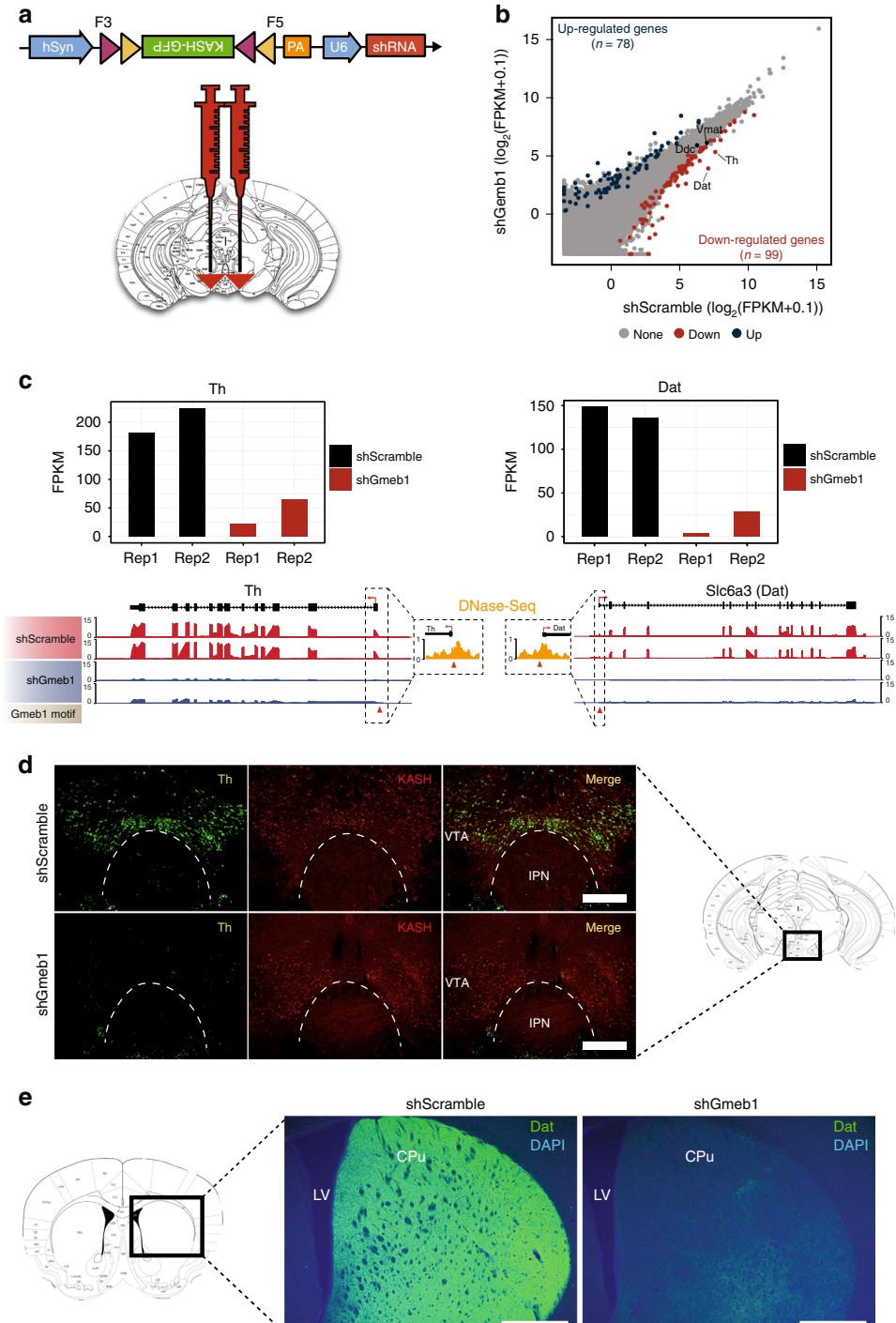

**Fig. 3** Midbrain *Gmeb1* knockdown results in loss of Th and Dat. **a** Diagram of the DIO-KASH-GFP-shRNA expressing vector (top) and location of midbrain virus injection in Dat-Cre mice (bottom). **b** Scatter plot comparing RNA-Seq data of mDA neurons of control (shScramble, *n* = 2) and *Gmeb1* knockdown (shGmeb1, *n* = 2). *Gmeb1* knockdown results in down-regulation of 99 genes (blue), including *Th* and *Dat*, and up-regulation of 78 genes (red) (fold change > 2; *p-value* < 0.05). **c** Histograms (RPKM) and genome browser view showing down-regulation of *Th* and *Dat* following *Gmeb1* knockdown. Dashed inserts: Location of *Gmeb1* motifs in promoter DHS for *Th* and *Dat* (red triangles). **d** Representative immunostaining pictures showing KASH-HA tagged mDA nuclei (red) and reduction of Th (green) in VTA of shGmeb1-treated mice. Box inserts reflect brain region where micrograph was obtained. Scale bar: 500 μm. Note: Coronal brain section was reproduced from ref. [48]. (Copyright 2013, Elsevier, Academic Press). **e** Representative immunostaining pictures showing reduction of Dat (green) in caudate putamen (CPu) of shGmeb1-treated mice. LV; Lateral ventricle. Box inserts reflect brain region where micrograph was obtained. Scale bar: 500 μm. Note: Coronal brain section was reproduced from ref. [48]. (Copyright 2013, Elsevier, Academic Press)

proficiency, *Gmeb1* knockdown mice showed almost complete inability to swim (Fig. 5c, *n* = 11/group, ****p* < 0.0001 with two-tailed *t*-test). In contrast, when the shScramble mice were placed in the water-filled chamber, they adapted their motor repertoire and swam for at least 5 min while *Gmeb1* knockdown mice generally floated with their hind limbs spread horizontally and struggled to coordinate the hind limb movement necessary to swim (Supplementary Movie 3). These results suggest that *Gmeb1*

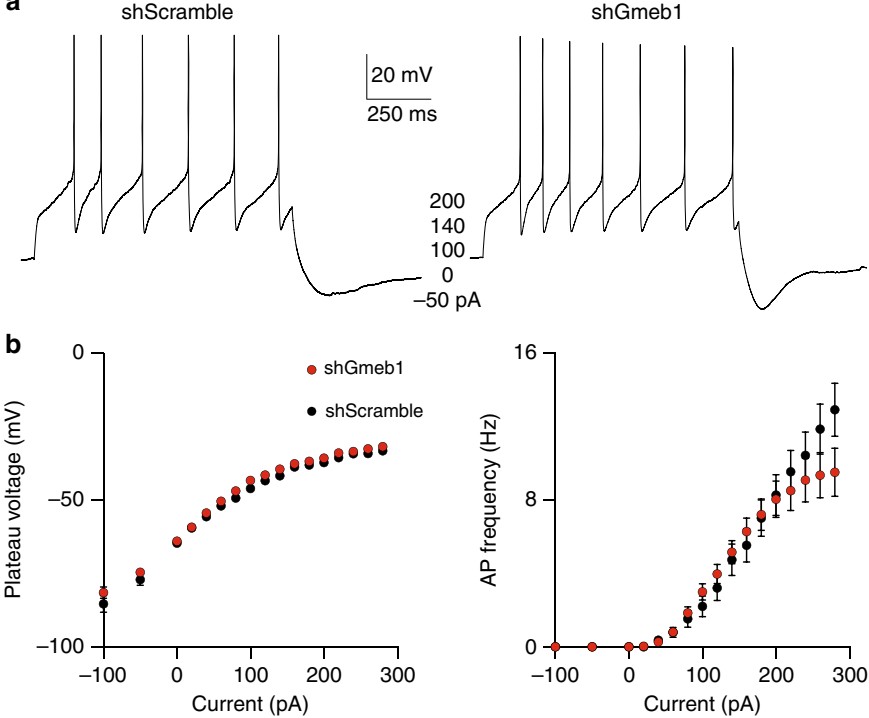

**Fig. 4** *Gmeb1* knockdown does not impair electrophysiological function in SNc mDA neurons. **a** Sample current-clamp recordings of action potentials from SNc mDA neurons following injection of depolarizing currents (pA). **b** *Gmeb1* knockdown does not affect the mean plateau voltage or AP frequency of SNc mDA neurons. Shown are the mean (±s.e.m.) plateau voltage (mV) curves (left) and the mean (±s.e.m.) action potential frequency (Hz) (right) of *Gmeb1* knockdown (*n* = 26 cells) or control (*n* = 19 cells) SNc mDA neurons following incremental current (pA) injection (right)

knockdown results in motor deficits, especially when animals are challenged with a situation that requires a new form of motor response.

Lastly, to determine if muscle endurance contributed to the above *Gmeb1* loss-of-function phenotypes, we subjected the mouse cohorts to the hanging wire test. In this test, mice grasp a 2-mm diameter steel wire 40 cm above a padded surface so that the animal hangs by gripping the wire with its forepaws. The time (latency), required for the mouse to fall is recorded, where lower latency to fall indicates lower muscle endurance. We found no significant difference between the groups (Fig. 5d, Supplementary Movie 4, *n* = 11/group, *p* = 0.3462 with two-tailed *t*-test), suggesting that muscle endurance is not affected, and that the effects seen in the pole test, rotarod and swim tests are most likely due to deficits in balance and/or coordination. While muscle endurance is compromised in PD, this phenomenon can also be attributed in part to muscle atrophy[27]. Given that mice were tested approximately two weeks after virus injection, it is not surprising that this aspect of motor control was not yet affected by *Gmeb1* knockdown. Further, knocking down *Gmeb1* did not result in hypolocomotion, as measured by distance traveled (Supplementary Fig. 12). Taken together, while we cannot completely rule out the possibility that the defects on balance and coordination following Gmeb1 knockdown in SNc are due to non-mDA effects, these behavioral results are consistent with pharmacological models of dopamine depletion[24,25], suggesting that the phenotypes are likely due to downregulation of *Th*.

## Discussion
Understanding transcriptional regulation of genetically-defined neuron populations in vivo has proven difficult due to the cellular heterogeneity of the mammalian brain. Despite efforts in profiling the mDA transcriptome using tools such as TRAP-Seq[28,29] and single-cell RNA-Seq[30], these approaches do not allow for chromatin analysis, and therefore do not provide mechanistic insights into how mDA genes are regulated—especially those directly involved in dopamine signaling. To overcome these technical hurdles, we developed an in vivo virus-based approach to tag and purify mDA nuclei for transcriptome and chromatin accessibility analysis (Fig. 1a).

mDA neurons play a central role in reward signaling and are widely implicated in psychiatric and neurodegenerative disorders[2,3]. Therefore, a better understanding of transcriptional regulation in this neuronal population is critical. To our knowledge, this is the first study to generate a genome-wide chromatin accessibility map of mDA neurons. By comparing transcriptome and accessible chromatin maps of mDA neurons to those of cortical neurons, we identified candidate TFs predicted to regulate mDA-enriched gene expression (Fig. 2d–f). While our TF motif analyses identified known mDA transcriptional regulators, such as *Nurr1* and *Foxa2*[18,19] (Fig. 2e), it also uncovered novel transcriptional regulators, such as *Gmeb1*, whose function in mDA neurons was previously unknown.

As a proof-of-principle study, we demonstrated that *Gmeb1* plays an important role in regulating mDA-enriched genes, such as *Th* and *Dat*, which are essential for dopamine signaling (Fig. 3). Indeed, *Gmeb1* knockdown reduces *Th* and *Dat* expression, yet it does not affect mDA neuron survival (Supplementary Figs. 9, 10) or basic electrophysiological functions (Fig. 4). This suggests that while *Gmeb1* knockdown may not affect the ability of mDA neurons to communicate, due to the essential role of Th in dopamine synthesis, their reduced Th levels may compromise homeostatic dopamine signaling. Consistent with this notion, knockdown of *Gmeb1* in the SNc of adult mice resulted in motor deficits, similar to pharmacological models of PD[25] (Fig. 5). Collectively, our study reveals *Gmeb1* as a novel transcriptional regulator essential for mDA neuron function.

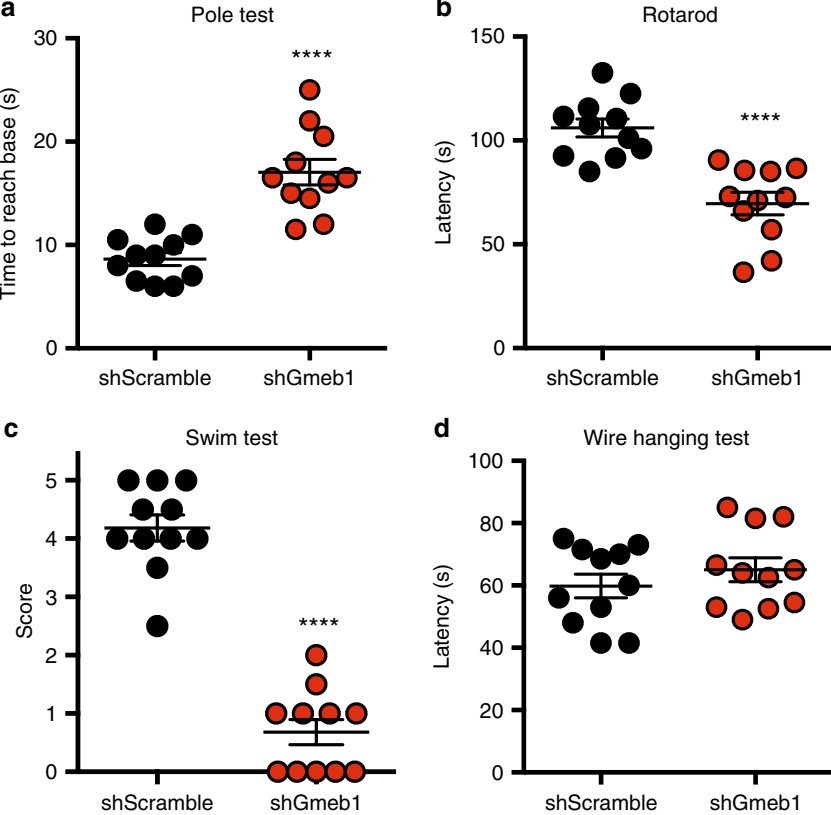

**Fig. 5** Midbrain *Gmeb1* knockdown results in motor impairments. **a** Dot plot representing mean (±s.e.m.) time (s) required for mice with midbrain knockdown of *Gmeb1* or shScramble control mice to reach bottom of pole ($n = 11$ per group, ****$p < 0.0001$ in two-tailed *t*-test). **b** Dot plot representing mean (±s.e.m.) latency (s) at which mice with midbrain knockdown of *Gmeb1* or shScramble control mice fall from rotarod or cling to apparatus without active locomotion ($n = 11$ per group, ****$p < 0.0001$ in two-tailed *t*-test). **c** Dot plot representing mean (±s.e.m.) score reflecting the ability of *Gmeb1* or shScramble to swim over a 5-min period ($n = 11$ per group, ****$p < 0.0001$ in two-tailed *t*-test). **d** Dot plot representing mean (±s.e.m.) latency (s) at which mice with midbrain knockdown of *Gmeb1* or shScramble control mice fall from horizontal wire ($n = 11$ per group, $p = 0.3462$ in two-tailed *t*-test)

A recent report has shown that it is possible to infer cell type-specific transcriptional regulators by using single-cell ATAC-Seq to catalog accessible chromatin regions from heterogeneous mouse forebrain tissue[6]. Indeed, identifying *Gmeb1* would not have been possible without generating an mDA-specific chromatin accessibility map. Therefore, reporting the role of *Gmeb1* in regulating the expression of mDA identity genes and in maintaining homeostatic motor control highlights the potential of combining cell-type specific chromatin accessibility profiling with RNA-Seq to identify novel transcriptional regulators. Further, this finding raises the possibility that other neuron populations in brain may be similarly regulated by otherwise unknown TFs. To this point, our nuclear tagging approach could be applicable to any Cre-expressing mouse line, thereby eliminating the necessity of breeding multiple lines to target a single neuron type and thus reducing the time required to produce nuclei from genetically-defined cell populations. Since nuclei are isolated from mouse brain in its native form, this cell type-specific approach could also enable the study of chromatin and transcriptional changes associated with neurodegenerative disease, including higher-order brain functions such as reward learning and motivation.

## Methods

**Animals**. Female Dat-IRES-Cre heterozygous knock-in mice (Jackson Laboratories, 06660) were bred with male C57BL/6 J wildtype mice (Jackson Laboratories, 000664) to produce Dat-IRES-Cre heterozygous and wildtype offspring. Only male mice were used for profiling experiments whereas both male and female mice were used for histological analysis and establishment of the infection system. Mice were 8–12 weeks of age at the beginning of each experiment. All animal husbandry and

behavioral procedures were conducted in strict accordance with the NIH Guide for the Care and Use of Laboratory Animals and were approved by the Institutional Animal Care and Use Committee of Harvard Medical School.

**Genotyping**. Around 21 days of age, mouse pups were weaned and their ears were clipped for genetic analysis. Genomic DNA was extracted using the hot sodium hydroxide and tris (HotSHOT) method[31]. Primers for the Dat wildtype and Dat-Cre mutant genes were: oIMR6625—common (5′-TGGCTGTTGGTGTAAAGT GG-3′); oIMR6626—wildtype reverse (5′-GGACAGGGACATGGTTGACT-3′), DAT-Cre–mutant reverse (5′-CCAAAAGACGGCAATATGGT-3′). Samples were processed for genetic amplification with PCR and analyzed on a 1.5% agarose gel with GelRed Nucleic Acid Stain. The band for the DAT-Cre wildtype gene was at 264 bp, and the Dat-Cre mutant gene was at 152 bp.

**Vector construction**. To generate the AAV-DIO-KASH-HA vector, KASH sequence (kind gift from Dr. Feng Zhang, Broad Institute) was cloned into the DIO cassette of cDIO-L10-VHH-HA (kind gift from Dr. Jeffrey Friedman, Rockefeller University), replacing L10-VHH sequence. To generate the AAV-DIO-KASH-GFP-U6-shRNA vector, the EGFP sequence of the AAV-hsyn-cDIO-EGFP vector (Addgene 50457) was replaced by KASH-GFP sequence and then the U6-shRNA sequence was inserted after the polyA sequence. To target mouse *Gmeb1*, 5′-AT TACTCCTGTGGGCCAGTCC-3′ shRNA sequence was used (sh1). Other *Gmeb1* shRNA sequences tested were 5′-CTAAAACTCAAGTGATCTTGC-3′ (sh2) and 5′-TTTACCAGCATGACAAAGTTT-3′ (sh3). The scramble sequence 5′-CCTAAGGTTAAGTCGCCCTCG-3′ was used as a control. All plasmids were expressed in *E. coli* and packaged into AAV serotype 5 by the Boston Children's Hospital Viral Vector Core. AAV-hSyn-DIO-mCherry virus was purchased from Addgene (50459-AAV5).

**Stereotaxic surgery and viral delivery**. Animals were anesthetized with a 1–3% isolflurane/oxygen mixture and mounted in a stereotaxic frame (Kopf Instruments, Tujunga, CA) at a "flat-skull" position. Using aseptic technique, a 5 mm longitudinal incision was made on the skin overlying the skull, exposing Bregma. Two

small circular openings were drilled on the skull to expose the dura surface overlying the midbrain. Two bilateral injections (0.5 µl each at a flow rate of 0.2 µl per min) were made at the following coordinates: For midbrain, anterior-posterior (AP) −2.95 mm; medial-lateral (ML): ± 0.5 mm from midline; dorsal-ventral (DV) −4.2 mm from dura. For SNc, AP −2.95 mm; ML ± 1.4 mm from midline; DV −4.3 mm from dura. To ensure proper viral dispersion throughout midbrain parenchyma, the 30-gauge needle was left in place for 5 min before retracting.

**Tissue dissection, brain perfusion, and fixation**. For RNA-Seq and liDNase-Seq, mice were euthanized by inhalation of $CO_2$. Brains were rapidly removed and midbrain was dissected with a scalpel. Samples were immediately frozen in liquid nitrogen and then stored at −80 °C until processing for nuclear isolation. For histological analysis, mice were euthanized by inhalation of $CO_2$ and immediately perfused through the ascending aorta with 0.9% saline, followed by 4% paraformaldehyde (PFA) in 0.1 M phosphate buffer saline (PBS; pH 7.4). Brains were harvested, postfixed overnight in 4% PFA, and then stored in 30% sucrose in 0.1 M phosphate buffer (pH 7.4) for 72 h. All brains were cut into 40 µm coronal sections on a cryostat, and the floating sections were stored in 0.1 M PBS with 0.01% sodium azide at 4 °C until processing for fluorescence immunolabeling.

**Nuclear isolation**. Frozen midbrain samples were homogenized in 1 ml ice-cold homogenization buffer [320 mM sucrose, 5 mM $CaCl_2$, 3 mM $Mg(Ac)_2$, 10 mM Tris pH7.6, 0.1 mM EDTA, 0.1% NP40, 0.1 mM PMSF, 1 mM β-mercaptoethanol, 1% BSA, 1:250 RNasin Plus RNase Inhibitor (Clontech)] using a 1 ml Dounce homogenizer (Wheaton); 30 times with pestle A, followed by 30 times with pestle B. After 10 min on ice, the homogenate was filtered with 40 µm cell strainer (Fisher) and added 1 ml dilution buffer [50% OptiPrep density gradient medium (Sigma), 5 mM $CaCl_2$, 3 mM $Mg(Ac)_2$, 10 mM Tris pH 7.6, 0.1 mM PMSF, 1 mM β-mercaptoethanol] and mixed thoroughly with pipette. Loaded 0.5 ml lysate on the top of 0.5 ml 29% iso-osmolar OptiPrep solution (in PBS) in a 1.5 ml centrifuge tube and centrifuged at 6000×g for 10 min at 4 °C. After removing the supernatant, the nuclei were resuspended in wash buffer [2.5 mM $MgCl_2$, 1% BSA in PBS, 1:500 RNasin Plus RNase Inhibitor (Clontech)] for immunostaining or FANS.

**Nuclear immunolabeling**. Isolated nuclei suspended in wash buffer (1% BSA, 2.5 mM $MgCl_2$ in PBS) were centrifuged (5 min at 1000×g) and supernatant was discarded. Nuclear pellet was then resuspended with rabbit anti-HA (1:200, 3724, Cell Signaling) or chicken anti-GFP (1:2000, Ab-13970, Abcam) in 200 µl wash buffer for 1 h and centrifuged (5 min at 1000×g). Supernatant was discarded and pellet was resuspended in 400 µl wash buffer for 5 mins and centrifuged again (5 mins at 1000×g). Nuclear pellet was then resuspended in donkey anti-rabbit 568 (1:500; A10042, Invitrogen) or TRITC-conjugated donkey anti-chicken secondary (1:500, 703–025–155, Jackson Immunoresearch) in 200 µl wash buffer for 1 h and then centrifuged (5 mins at 1000×g). Supernatant was discarded and pellet was resuspended in 400 µl wash buffer for 5 mins and centrifuged again (5 mins at 1000×g). Nuclei were resuspended with 1% DAPI in 300 µl wash buffer. All steps for nuclear staining were performed at 4 °C.

**Fluorescence immunolabeling**. Floating sections (40 µm thickness) were processed for fluorescent immunostaining of HA, and TH. Sections were rinsed in 0.1 M PBS, pH 7.4, with 0.5% Triton-X 100 (PBT) and then blocked in 10% normal donkey serum/PBT for 1 h. Then, sections were incubated in the primary antibody in PBT at 4 °C overnight. The primary antibodies were diluted as follows: mouse anti-Th (1:500, SC-25269, Santa Cruz), rabbit anti-HA (1:800, 3724, Cell Signaling), Chicken anti-GFP (1:2500, Ab-13970, Abcam) and rat anti-Dat (1:500, GTX30992, GeneTex), Rabbit anti-Caspase-3 (1:500, Ab-13847, Abcam). The following day, the sections were rinsed and incubated in two of the following secondary antibodies: Alexa 568 donkey anti-rabbit (A10042, Invitrogen), Alexa 488 donkey anti-mouse (A21202, Invitrogen), FITC-conjugated donkey anti-chicken (703-095-155, Jackson Immunoresearch), TRITC-conjugated donkey anti-chicken (703-025-155, Jackson Immunoresearch), and Alexa 488 donkey anti-rat (A21208, Invitrogen) (all secondary antibodies were used at 1:500 concentration). Sections were incubated with secondary antibodies in PBS (in 2% donkey serum) for 2 h. Next, the sections were rinsed, mounted on slides with VectaShield (with DAPI) (H-1200, Vector Labs), and coverslipped. Controls included processing the secondary antibodies alone to verify background staining, processing each primary with the secondary antibody to verify channel-specific excitation, examining for autofluorescence in an alternate channel with tissue lacking that channel-specific probe. Only the brightness and/or contrast levels were adjusted post-acquisition and were imposed across the entire image.

**TUNEL assay**. To analyze whether knocking down Gmeb1 would affect mDA neuron survival, 20 µm midbrain slices from Dat-Cre mice injected with either a 1:1 mixture of shGmeb1 virus and DIO-mCherry virus, or shScramble virus and DIO-mCherry virus. Midbrain slices were first immunolabeled with rabbit anti-mCherry (1:500, ab167453, Abcam) and mouse anti-Th (1:500, SC-25269, Santa Cruz) antibodies, followed by TUNEL assay, using the Click-iT Plus TUNEL Assay for In Situ Apoptosis Detection kit-Alexa Fluor 647 dye (Thermo Fisher C10619).

As a positive control, midbrain slices were pre-treated with DNase-1 (1 unit/50 µl), which generated positive TUNEL signals by inducing DNA double strand breaks.

**Electrophysiology**. Eight week-old Dat-Cre heterozygous male received SNc injections of a 1:1 virus mixture, with one side receiving a control mix (DIO-mCherry/shScramble; 0.5 µl total volume) and the contralateral SNc receiving the knockdown mix (DIO-mCherry/shGmeb1; 0.5 µl total volume). Two weeks following injections, brains were collected and 300 µm coronal slices were cut in ice-cold artificial cerebrospinal fluid (ACSF) containing (in mM) 125 NaCl, 2.5 KCl, 25 $NaHCO_3$, 2 $CaCl_2$, 1 $MgCl_2$, 1.25 $NaH_2PO_4$, and 25 glucose (295 mOsm/kg). Slices were transferred and treated for 10 min in a chamber at 32 °C containing choline-based solution consisting of (in mM): 110 choline chloride, 25 NaHCO3, 2.5 KCl, 7 $MgCl_2$, 0.5 $CaCl_2$, 1.25 NaH2PO4, 25 glucose, 11.6 ascorbic acid, and 3.1 pyruvic acid before transferring to a second chamber with ACSF at 32 °C for at least 30 min. Recordings were performed at 32 °C in carbogen bubbled ACSF. K-based internals for current-clamp to measure neuron intrinsic properties (in mM: 135 $KMeSO_3$, 3 KCl, 10 HEPES, 1 EGTA, 0.1 $CaCl_2$, 4 Mg-ATP, 0.3 Na-GTP, 8 $Na_2$-Phosphocreatine, pH 7.3 adjusted with KOH; 295 mOsm·kg$^{-1}$). Under current-clamp, we maintained membrane potential at −65 mV by injecting current, then increased current by incremental steps, starting from −100 to 300 pA. Data obtained were analyzed using the parameters described in Wallace et al.[32].

**RNA-Seq**. FANS-sorted nuclei were directly sorted to RLT plus buffer (Qiagen). Total RNA and DNA were prepared from the same samples. Total RNA samples from 400 nuclei were reverse transcribed and amplified using SMARTer Ultra Low Input RNA cDNA preparation kit (Clontech). cDNAs were then fragmented using the Covaris M220 sonicator (Covaris). The fragmented cDNAs were end-repaired, adaptor ligated and amplified using NEBNext Ultra DNA Library Prep Kit for Illumina according to the manufacturer's instruction (New England Biolabs). Single end 100 bp sequencing was performed on a HiSeq2500 sequencer (Illumina).

**Low-input DNase-Seq**. Low-input DNase-Seq (liDNase-Seq) libraries were prepared as previously described[12]. Nuclei were collected in 1.5 ml tubes in PBS with BSA during sorting. A total of 500 nuclei were pelleted and resuspended in 36 µl lysis buffer (10 mM Tris-HCl, pH 7.5, 10 mM NaCl, 3 mM $MgCl_2$, 0.1% Triton X-100) and incubated on ice for 5 min. DNase I (10 U/µl, Roche) was added to the final concentration of 40 U/ml and incubated at 37 °C for 5 min. The reaction was stopped by adding 80 µl Stop Buffer (10 mM Tris-HCl, pH 7.5, 10 mM NaCl, 0.15% SDS, 10 mM EDTA) containing 2 µl Proteinase K (20 mg/ml, Life technologies). Then 20 ng of a circular carrier DNA [a pure plasmid DNA without any mammalian genes was purified with 0.5× Beckman SPRIselect beads (Beckman Coulter) to remove small DNA fragments] was added. The mixture was incubated at 50 °C for 1 h, then DNA was purified by extraction with phenol-chloroform and precipitated by ethanol in the presence of linear acrylamide (Life technologies) overnight at −20 °C. Precipitated DNA was resuspended in 50 µl TE (2.5 mM Tris, pH 7.6, 0.05 mM EDTA), and the entire volume was used for sequencing library construction.

Sequencing library was prepared using NEBNext Ultra DNA Library Prep Kit for Illumina (New England Biolabs) according to the manufactures' instruction with the exception that the adaptor ligation was performed with 0.03 µM adaptor in the ligation reaction for 30 min at 20 °C, and that PCR amplification was performed using Kapa Hifi hotstart readymix (Kapa Biosystems) for 8-cycles. The PCR products were purified with ×1.3 volume of SPRIselect beads (Beckman Coulter) and then size selected with ×0.65 volume followed by ×0.7 volume of SPRIselect beads. The samples were amplified again with Kapa Hifi hotstart readymix. The PCR product was purified with ×1.3 volume of SPRIselect beads and then size selected with ×0.65 volume followed by ×0.7 volume of SPRIselect beads. The DNA was eluted in TE. The libraries were sequenced on a Hiseq2500 with single-end 100 bp reads (Illumina).

**RNA-Seq analysis**. All RNA-Seq data used in this study (produced by our lab or from public datasets), were mapped to the mm9 genome. Prior to mapping, raw RNA-Seq datasets were first trimmed using Trimmomatic[33] (v.0.36). Illumina sequence adaptors were removed, the leading and tailing low-quality base-pairs (less than 3) were trimmed, and a 4-bp sliding window was used to scan the reads and trim when the window mean quality dropped below 15. Next, reads with a length of at least 50-bp were mapped to the genome using STAR[34] (v.2.5.2b) with the following parameters: –outSAMtype BAM SortedByCoordinate –outSAMunmapped Within –outFilterType BySJout–outSAMattributes NH HI AS NM MD–outFilterMultimapNmax 20–outFilterMismatchNmax 999. The resulting bam files were then passed to RSEM[35] to estimate genes and isoform abundance given the Ensemble NCBIM37.67 transcriptome assembly.

**Differential gene expression analysis**. The R/Bioconductor DESeq2[36] package was used to detect the differentially expressed genes between purified mDA neurons and public Exc, PV and VIP samples[5]. Genes showing more than four-fold expression change and a q-value < 0.001 were considered as differentially expressed. In the case where only a single replicate was available (HA$^+$ vs. HA$^−$), only genes with an expression level of FPKM > 1 and that showed at least a four-fold

expression change were considered. For the *Gmeb1* knockdown RNA-Seq data, a lower cutoff criterion (FPKM > 1, *q*-value < 0.05 and FC > 2) was used due to the variability between the knockdown samples.

**Functional enrichment analysis.** The enrichGO and enrichKEGG function from the R/Bioconductor clusterProfiler[37] package were, respectively, used to perform GO and pathway enrichment analysis. Only the GO terms and pathways with an adjusted *p*-value < 0.05 were considered. The associated GO and pathway enrichment plots were generated using the ggplot2[38] package.

**Plot generation.** Heatmaps were generated using the R/Bioconductor package ComplexHeatmap[39]. Boxplots were generated using the standard R boxplot function. All the other plots were generated using the ggplot2 package.

**DNase-Seq data analysis.** Raw 100-bp single-end DNaseq-Seq reads were first trimmed with Trimmomatic[33] (v.0.36) with parameters similar to those used in RNA-Seq, except that trimmed reads were allowed to have a length of at least 25-bp. Next, we used Bowtie2[40] to map the trimmed reads to the mm9 genome without allowing any mismatches. Unmapped reads and reads with mapping quality less than 10 were filtered out using samtools[35] (v.1.3.1). DNaseq-Seq peaks were then called using macs2[41] with the parameters–broad–nomodel–nolambda. Peaks with a *q*-value < 0.05 were considered for further analysis.

**ATAC-Seq analysis.** ATAC-Seq reads for each replicate of Exc, PV and VIP neurons[5] were mapped to the mm9 genome. Raw reads were first trimmed using Trimmomatic with the same parameters used for liDNase-Seq analysis. The trimmed pair-end reads were then mapped using Bowtie2 with parameter-X 2000. Unmapped reads, reads with mapping quality less than 10 and reads mapping to the mitochondrial genome were discarded. Macs2 was then used with parameters–broad–nomodel–nolambda to call peaks. Peaks with *q*-value < 0.05 were considered.

**Peaks genomic annotation.** NCBIM37.67 was used as a source of gene annotation. Mainly, the annotatePeak function from the R/Bioconductor ChIPpeakAnno[42] package was used for genomic annotation. Promoters were defined by ±3kb from TSS of the transcripts, and all regions that did not fall within exons, introns, or UTRs were classified as distal intergenic regions.

**Promoter DHS motif analysis.** Motif analysis was performed using the R/Bioconductor PWMEnrich package[43]. To find motifs enriched at the promoters of mDA-enriched genes, we first constructed a lognormal background motif distribution of the 200-bp chunks generated from the TSS ± 3 kb of all the annotated NCBIM37.67 mouse (mm9) genes. As a foreground, we used the mDA DHS peaks located within 3 kb from the TSS of the 107 mDA-enriched genes. A total of 59 genes out of 107 mDA-enriched genes contained a promoter DHS. The motifEnrichment function was used to detect the most enriched motifs.

**Distal DHS motif analysis.** To locate distal regulators of mDA neurons, we selected DHSs located 3 kb to 1 Mb away from TSS for each of the 107 mDA-enriched genes. Next, we constructed a lognormal background motif distribution of the 200-bp chunks generated from the non-specific distal mDA DHS that overlapped with ATAC-Seq peaks in VIP, PV or Exc neurons. Then, the motifEnrichment function was used to find the enriched motifs. Because the most enriched distal motifs in our analysis were known mDA neurons regulators (Nurr1 and Foxa2), we adjusted the *p*-value to < 1e−2 and mRNA expression level to FPKM > 1 to reveal potential new distal regulators.

**Associating promoter-enriched TF with their target genes.** Motif analysis only produced a list of over-represented TF binding-site (TFBS) in promoter DHS, but did not directly indicate which TF would regulate which gene(s). To predict the potential of a given TF to regulate a gene, we calculated a TF regulatory score (TFRS) for each TF-gene pair. Initially, we defined four regression models (elastic net regression) by finding a correlation between gene expression and different TF binding features, then, we selected the model with the best predictive outcome. Three features were considered: (i) the binding affinity of the TF to the gene's promoter DHS (represented by the TF motif *p-value*), (ii) the strength of the gene's promoter DHS peak signal (RPKM) and (iii) the distance of the promoter DHS peak to the gene's TSS.

For each model, the 59-gene by 11-TF matrix was constructed in which the TFRS score of $TF_i$ to a gene $g$ was calculated with one of the following model scores:

$$\text{TFRS}_{ig}^{pval} = -\log_{10}\left(p\text{value}_{ig}\right) \qquad (1)$$

$$\text{TFRS}_{ig}^{pval\_DHS} = -\log_{10}\left(p\text{value}_{ig}\right) \times \text{RPKM}_{\text{DHS}(g)} \qquad (2)$$

$$\text{TFRS}_{ig}^{pval\_dist} = -\log_{10}\left(p\text{value}_{ig}\right) \times \exp\left(-d_{ig}/100\right) \qquad (3)$$

$$\text{TFRS}_{ig}^{pval\_DHS\_dist} = -\log_{10}\left(p\text{value}_{ig}\right) \times \text{RPKM}_{\text{DHS}(g)} \times \exp(-d_{ig}/100) \qquad (4)$$

Where: $\text{RPKM}_{\text{DHS}(g)}$ is the liDNase-Seq RPKM value of the promoter DHS of gene $g$, and $p\text{value}_{ig}$ is the enrichment *p*-value of the $TF_i$ motifs at the promoter DHS of gene $g$. $d_{ig}$ is the distance (in bp) between the promoter DHS and the gene $g$.

For each model, we used the caret package to train an elastic net regression from the glmnet package, with the following objective function:

$$\min_{(\beta_0,\beta)\in\mathbb{R}^{p+1}} \frac{1}{2N}\sum_{g=1}^{N}\left(y_i - \beta_0 - x_g^T\beta\right)^2 + \lambda\left[(1-\alpha) \parallel \beta \parallel_2^2 + \alpha \parallel \beta \parallel_1\right] \qquad (5)$$

Where $\lambda \geq 0$ is the regularization parameter and $\alpha \in [0,1]$ is a tradeoff between ridge and lasso regression. The parameters were selected by varying $\lambda$ between 0 and 2 and $\alpha$ between 0 and 1 with a with a 0.1 step. $N = 59$ represents the number of genes and $\beta \in \mathbb{R}^p$ such as $p = 11$ (the number of TFs) are the coefficients to learn.

The performance of the models was evaluated using the root mean square error (RMSE) metric (Supplementary Fig. 6). Due to the limited number of genes (59), we trained the models using a 2-fold cross-validation. The model with the smallest RMSE was selected (in our case *p*-value + distance to TSS).

**Associating distal DHS enriched TFs with their target genes.** The TFs enriched in distal DHS have the potential to regulate the expression of genes from different distal open chromatin regions through chromatin loop formation. Thus, we assumed that the closer a TF binding site was to a TSS, the more regulatory effect this TF could have on the target gene. This assumption is based on the power-law decay observed for the chromatin loop formation probability calculated from Hi-C data[44]. Thus, we introduced an exponential decay penalty to account for this effect.

Given a $TF_i$ enriched at a distal DHS and a target gene $g$, the best TFRS scoring scheme was as follows:

$$\text{TFRS}_{ig} = \sum_{\text{dhs}_k \in \text{TSS}_g \pm 1\,\text{Mb}} -\log_{10}(p\text{value}_{ik}) \times \text{RPKM}_{\text{dhs}_k} \times \exp(-\frac{d_{ig}}{10\,\text{kb}}) \qquad (6)$$

Were $dhs_k$ is DHS located within a 1-Mb window around the TSS of gene $g$ and $d_{ig}$ is the distance between $TF_i$ and the gene $g$. $p\text{value}_{ik}$ is the enrichment *p*-value of the $TF_i$ motifs at $dhs_k$.

**Pipeline automation, statistical analysis, and genome visualization.** The RNA-Seq, ChIP-Seq, and ATAC-Seq mapping pipelines were automated using Bpipe[45]. All statistical and descriptive analyses were performed in the R environment (http://www.r-project.org/).

**Genomic tracks generation.** The generated RNA-Seq, DNaseq-Seq, and ATAC-Seq genomic tracks were scaled to a read-per-million mapped reads genomeCoverageBed command from bedtools, then converted to bigwig format using the UCSC Genome Browser's bedGraphToBigWig utility. All genomic tracks visualization were performed using IGV[46]. Tracks of Fig. 3c were generated using GViz[47].

**Genomic datasets.** All the datasets generated in this study are described in Supplementary Data 8. The RNA-Seq and ATAC-Seq datasets for VIP, PV and excitatory neurons analyzed in this study were all from GSE63137[5].

**Pole test.** C57BL/6 mice with either shGmeb1 or shScramble virus injection were subjected to the pole test. Mice were placed at the top of a 50 cm grooved metal pole with a diameter of 1 cm with the head pointing down. The time from initial placement on the top of the pole to the time the mouse reaches the base of the pole (forelimb contact with platform) was recorded with a stopwatch, representing the pole test score (seconds). Following a 30-min rest period in their home cage, the trial was repeated and both scores were then averaged to represent the composite score shown in Fig. 5a. Note: All behavior tests were performed approximately two weeks following shGmeb1 or shScramble virus injection.

**Rotarod test.** C57BL/6 mice with either shGmeb1 or shScramble virus injection were subjected to the rotarod test. Each mouse underwent a habituation period, followed by two test trials. For habituation, mice were placed on the rotarod apparatus (Ugo Basile SRL, Varese, Italy) for 5 min at a speed of 4 RPM and then returned to their home cage for 10 min. Number of falls during the habituation period was recorded. For the testing session, mice were again placed in the rotarod apparatus rotating at an initial speed of 4 RPM. The speed gradually increased from 4 RPM to 40 RPM over the span of 3 min, and remained at 40 RPM for the final 2 min of the 5-min testing session. The time at which the mouse either fell from the rotarod or grasped on to the rotarod for 1 revolution without trotting was recorded and termed as the "latency" score. Once the latency score was obtained, the testing

session ended and the mouse returned to its home cage. The testing session was repeated 30 min later. Scores for each session were averaged to represent the composite score shown in Fig. 5b.

**Swim test**. C57BL/6 mice with either shGmeb1 or shScramble virus injection were subjected to the swim test. Mice were placed inside a 2 L beaker (13 cm diameter) filled with 1400 ml of water at 25 °C and were allowed to swim for 5 min, during which time the animals' swimming dexterity was scored. The swimming score (range 0–5) was assigned as follows: 5: continuous swimming (>80% of session); 4: continuous swimming (>60% of session) with occasional bouts of floating; 3: equal time (±10%) spent floating/swimming; 2: floating with occasional bouts of swimming (<20% of session); 1: Same as score of 2, but swimming bouts display limited hind limb paddling motion; 0: mice removed from the swim chamber before 5 min due to drowning. To prevent unnecessary distress, mice were removed from the chamber at the first sign of drowning. At the conclusion of the test, mice were removed, dried and returned to their home cage. Swim scores were averaged to represent the composite score shown in Fig. 5c.

**Hanging wire test**. C57BL/6 mice with either shGmeb1 or shScramble virus injection were subjected to the hanging wire test. Mice were placed on top of a steel wire 2 mm in diameter, suspended 40 cm above soft padding material. Placement of the mouse was such that both forepaws completely gripped the steel wire while the body hanged below. Once the mouse was placed on the steel wire, the time was recorded with a stopwatch until the mouse fell from the wire and landed on the soft padding surface. The amount of time the mouse hanged on the wire without falling was termed the "latency" score. The test was repeated twice with a 30-min interval between trials, during which the mouse was returned to its home cage. Latency scores from both trials were averaged to represent the composite score shown in Fig. 5d.

**Open-field locomotion**. C57BL/6 mice with either shGmeb1 or shScramble virus injection were subjected to open field locomotion test. Prior to testing, mice were habituated in the room for at least 30 min and then placed in the center of the open-field arena (Med Associates, ENV-510). The mouse was allowed to move freely in the arena for 30 min, which would be recorded as beam breaks and recorded as "distance traveled". At the conclusion of the test, the mouse was removed and returned to its home cage. Distance traveled (cm) scores for each group were averaged to represent the composite scores shown in Supplementary Fig. 10.

**Reporting summary**. Further information on research design is available in the Nature Research Reporting Summary linked to this article.

## Data availability
The accession number for the RNA-seq and liDNase-seq data presented in this study is available from the Gene Expression Omnibus (GEO) database under accession GSE106956.

## Code availability
Additional custom codes used for bioinformatics analysis are available upon request.

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

## Acknowledgements

We thank Drs. Jeffrey Friedman (Rockefeller U) and Feng Zhang (Broad Institute) for providing plasmids; Drs. Wenqiang Chen and Aritra Bhattacherjee for critical reading of the manuscript; Dr. Barbara Caldarone and Paul Lorello (Harvard Neurobehavior Core) for providing logistical assistance with animal studies, and the reviewers of this manuscript for their time and constructive feedback. This project is supported by NIH R01DA042283 (to Y.Z.), NIH K01DA045294 (to L.M.T.) and a Klarman Family Foundation Eating Disorders Research grant (to Y.Z.). Y.Z. is an investigator of the Howard Hughes Medical Institute.

## Author contributions

Y.Z. conceived the project; L.M.T., B.L.S., and Y.Z. designed the experiments and wrote the manuscript; L.M.T., M.N.D., R.C., F.L., W.W., and B.L.S. performed experiments and analyzed the data.

## Additional information

**Competing interests:** The authors declare no competing interests.

**Journal Peer Review Information:** *Nature Communications* thanks Angel Barco, and the other, anonymous, reviewer(s) for their contribution to the peer review of this work. Peer reviewer reports are available.

