## [Peer Review File · Nature Communications]

Reviewers' comments:

Reviewer #1 (Remarks to the Author):

In the manuscript, entitled "Gmeb1 is a transcriptional regulator important for dopamine neuron function," Tuesta, Zhang and colleagues present a series of compelling data—using sophisticated molecular tools and approaches—to demonstrate a potentially novel role for the Glucocorticoid modulatory element-binding protein 1 in the regulation of dopaminergic gene expression, and thus motor function, in brain. To do so, the authors first expressed an AAV5-DIO-KASH-HA virus into the midbrain (hitting both VTA and SNc) of DAT-cre mice, followed by HA associated FANS purifications, RNA-seq and low input DNase-seq to obtain gene expression and chromatin accessibility profiles specifically within DA neurons (vs. non-HA+ cells - mainly glial cells based on the seq data). In doing so, the authors provide very nice controls demonstrating the validity of this approach including comparisons of alternative neuronal subtypes in the cortex (e.g., VIP, Exc and PV neurons). Following identification of DAergic enriched gene expression and DHS, motif analyses were employed in an attempt to identify putative upstream TFs that may be important for mediating DAergic specific gene expression patterns; such assessments resulted in the identification of Gmeb1, alongside with other well known regulators of DAergic gene expression (e.g., Atf1 and Creb1), as a possible upstream regulator of DAergic genes, such as Th and Slc18a2. To next assess whether Gmeb1 itself is truly able to mediate these patterns of gene expression, conditional shRNA vectors targeting Gmeb1 were injected into the midbrain of DAT-Cre mice, followed again by FANS RNA-seq, as well as immunohistochemical validations of Dat expression in the dorsal striatum. Finally, to demonstrate that knocking down Gmeb1 in DAergic neurons, which reduces both Th and Dat expression, actually leads to functional consequences, the authors performed numerous assessments of motor outputs including Swim and Pole tests, Rotarod and the Hanging Wire test - in doing so, they validate that DAergic Gmeb1 KD does indeed lead to predicted deficits in striatal driven motor functions. Overall, this is a thorough, well designed and interesting manuscript that would be of broad interest to the field. However, there are a few points that need to be addressed prior to publication in Nature Communications:

1) Perhaps I missed it, but is Gmeb1 preferentially enriched in DAergic neurons vs. the other neuronal subtypes assessed (or even vs. glia for that matter)? If so, then the overall claims of the manuscript seem reasonable; however, if this TF is more broadly expressed in brain (as would be suggested from the Allen Brain Atlas), then the authors should tone down their claims relating to the specificity of Gmeb1 activity in the mediation of DAergic gene expression, per se.

2) While I greatly appreciate the numerous controls provided for the Gmeb1 KD studies, it would have been nice to also see a Gmeb1 overexpression experiment to demonstrate bidirectional effects of Gmeb1 manipulations on the expression of target genes (either by RNA-seq, or even with just a focused analysis of Th and Dat).

3) Is Gmeb1's regulation of genes, such as Th and Dat, more of a homeostatic regulatory mechanism in adult brain, or do the authors envision that Gmeb1 is also responsible for activity-dependent regulation of these genes? In other words, it is surprising to me that KD of Gmeb1 alone leads to such dramatic decreases in Th and Dat expression, while other TFs, such as Atf1 and Creb1, are also thought to be involved in regulating their expression. Therefore, I am wondering if Gmeb1 may play more of a homeostatic maintenance function, while TFs like Creb1 may be more important for activity-dependent regulation. This could be tested by giving mice drugs (e.g., cocaine or amphetamine) or optogenetic/chemogenetic stimulations, that increase DA release from VTA into striatum—manipulations that also tend to lead to increased expression of Th and/or Dat in VTA—in the context Gmeb1 KD to see if stimulation induced gene expression of these genes is similarly controlled by Gmeb1 (or if these patterns of expression are more controlled by alternative TFs).

4) Given the profound impact of Gmeb1 KD on Dat expression in mice (i.e., it looks to be almost completely ablated), I am curious whether the authors have performed measurements of either DA levels in striatum and/or firing activity of VTA DAergic neurons in the context of Gmeb1? With such reductions in gene expression and motor functions, are the author sure that the infected neurons remain healthy with Gmeb1 KD? This would be an important control for both downstream gene expression and behavioral studies.

5) Finally, given the apparent critical importance of Gmeb1 expression in DAergic neurons in adulthood, I wonder if the authors have assessed its expression patterns or regulatory effects during DAergic neuronal development? For example, between E9 and E17 in mice, DAergic gene expression (including Th and Dat) are massively increased as DAergic neurons develop...is Gmeb1 important for these patterns of gene expression, or is it more involved in the maintenance of DAergic gene expression in adult animals?

In sum, this is a generally exciting piece of work, however, the inclusion of additional studies (as discussed above) aimed at nailing down Gmeb1's role in DAergic gene expression would greatly strengthen the manuscript.

Reviewer #2 (Remarks to the Author):

The manuscript by Tuesta and colleagues combines transcriptome and chromatin accessibility analyses to identify Gmeb1 as a new transcription factor regulating cell-specific gene expression in midbrain dopamine (mDA) neurons. Later, they use loss-of-function experiments (shRNA-based) to demonstrate that genes typically expressed in mDA cells, such as Th and Dat, are down-regulated, and that motor coordination results impaired.

The approach is technically innovative, interesting and could be applied to other discovery-driven projects in the nervous system. The follow-up proof-of-principle experiments are straightforward and support the notion that Gmeb1 is important for mDA function, although they do not delineate the actual role of this transcription factor in the biology of mDA cells. My specific criticisms are below:

1. The authors should check whether Gmeb1 loss-of-function affects neuronal viability. They should present histological analysis and explore the possible death of mDA neurons. The images presented in Sup. Fig. S8 represent a very compelling evidence for the downregulation of DA-related genes. The authors may consider including some of these images in a main figure along with higher magnification images showing if the cells that do not longer express Dat and Th are dying or remain viable.

2. To further differentiate if Gmeb1 is necessary for mDA cells viability or identity. The authors could consider including some electrophysiological experiments in neurons infected with AVV5-DIO-KASH-GFP-U6-shGmeb1. It would be very interesting to see whether these neurons have lost some mDA specific-features, but still behave as neurons (preserved resting potential, etc).

Minor issues:

3. Page 4: The description of GO enrichment analysis in the second paragraph should be revised. They indicate that differential profiling identified 107 genes; later they indicate that some genes were excluded from the list, and a few lines below they refer again to 107 genes. In which specific step and analyses were genes excluded? I should also note that the criterion for exclusion seems arbitrary and should be avoided in this type of analysis. GO analyses have some hypothesis-generating value, but should be conducted in an unbiased manner because the results of the analysis can be easily

influenced by ad hoc filtering of the gene list. For example, the authors claim in page 4 that they “defined these 107 genes as ‘mDA-enriched genes’”. Did they reach that conclusion after manually excluding genes that were “highly expressed in cortical neuron subtypes? This is not clear in the text.

4. Page 6: In the first paragraph the authors connect their genomic screen with the single gene study that occupies the second part of the manuscript. The reasons for focusing on Gmeb1 are unclear. They refer to a “predictive scheme accounted for the motif p-value and distance from DHS to TSS” that is explained in greater detail in Methods. Given the importance of this selection step, the authors should explain better the basis of their “predictive scheme” to a non-expert audience. The description in methods is very cryptic.

5. Page 8: The description of the authors suggests that swimming involves motor learning. However, swimming is an innate behavior in mice not a learned skill.

6. A Venn diagram does not seem the optimal representation for the results presented Figure 2a. As could only be expected, many intersection shows a 0 value. A sector graph seems more appropriate to present the percentage of genes unique for DA and those coincident with the different neuronal populations.

7. Introduction, 6 line: “...heterogeneity in (the) brain...”

Reviewer #3 (Remarks to the Author):

In this study, the authors use a novel viral-based approach to label mDA nuclei, sort them, and perform gene expression and chromosome accessibility analysis. They use previously published data on three cortical cell populations to define mDA-enriched genes and DHSs. Through this approach and predictive modeling to define TF binding sites for mDA-enriched genes, the authors discover a novel TF, Gmeb1, that appears to regulate expression of dopaminergic genes including Th and Slc6a3. They further show that in vivo knock-down of Gmeb1 reduces Th protein levels in the midbrain and striatum and causes motor impairments. While this study has some interesting components, further work is needed to clarify some key points, as described below.

Major points:

-An important limitation of this study is the way that mDA-enriched genes and DHSs were defined. These were defined by comparison to a previously published data set on three unrelated populations of cortical neurons. It is likely that if the authors used different cells as filters for exclusion (i.e. more closely related serotonergic or midbrain GABA-ergic neurons) that they would have a different list of genes/DHSs. In addition, it is unclear that Gmeb1 is uniquely required for DA neuron function – it could be highly active in other cell types that were not profiled. Also, by defining enriched genes in this way, the authors exclude genes that are very important for DA neuron development/function (e.g. Nurr1) but that happen to be expressed in one of the three cortical cell types.

-Related to the point above, it would be helpful if the authors could more clearly state their goal. Are they trying to define the TFs most important for DA neuron function? If so, then perhaps they should not filter genes based on expression in other cell types. Or, are the authors trying to identify TFs active in DA neurons exclusively? If this is the case, then a limited comparison to three cortical cell types does not rule out the possibility that the TFs they identify are active in other non-DA cells.

-The identification of Gmeb1 as a regulator of dopaminergic genes is interesting and novel. The authors perform proof-of-principle experiments that Gmeb1 knockdown affects expression of genes such as Th and causes motor impairments; however more information is needed to support their conclusions (see specific comments below).

Specific points:

- 1) What is the endogenous expression pattern of Gmeb1 in the midbrain? Is it expressed exclusively in DA neurons or also in surrounding astrocytes or non-DA neurons? Is it expressed equally in all DA neurons (i.e. VTA and SNc)? If it is expressed in non-DA cells, the observed phenotypes could be due to non-cell autonomous changes in other cell types that indirectly affect the health of the DA neurons. The authors could investigate Gmeb1 expression patterns with immunohistochemistry (if there is a good antibody) or in situ hybridization and cell-type specific markers.
- 2) Are the DA neurons dying with Gmeb1 knockdown? The authors could investigate this by crossing the DAT-Cre line to a tdTomato Cre reporter and/or examining markers of apoptosis at different time points post-injection.
- 3) Since it's possible that the DA neurons are simply sick/dying, the authors can't formally conclude that "loss of dopamine caused by Gmeb1 knockdown results in balance and coordination impairments" (pg 7) or "our results suggest that Gmeb1 knockdown in the SNc leads to motor impairments due to the disruption of dopamine signaling" (pg 8). If the authors want to claim that the motor deficits are due to loss of dopamine synthesis specifically, they should measure dopamine levels directly with HPLC or another method.
- 4) In Fig 3d, the authors only show an image of the VTA, but claim that the motor impairments are driven by changes in SNc DA neurons. In addition, it appears that the KASH-GFP is being expressed in non-DA cells (Th-negative) – if so, then this can't be used as a way to show that the DA neurons have not degenerated.
- 5) Are the Gmeb1 knock-down mice hypoactive? Multiple motor parameters including locomotor activity could be measured with a simple open field test.
- 6) To show that the motor impairments and gene expression changes are specific to Gmeb1 (and downregulation of genes controlling DA synthesis), the authors could perform injections of shRNA against another transcription factor, which is not expected to regulate Th expression (e.g. Tcf3 or Clock). This would help confirm the predictions made in Fig. 2f.
- 7) The authors should cite a relevant dopamine neuron single cell RNA-sequencing study in the discussion (pg 8) – La Manno et al, Cell, 2016. Reference #10 (Poulin et al, 2014) did not perform single-cell RNA-seq but rather used single-cell qPCR for 96 candidate genes.
- 8) What is the rationale for only focusing on promoter DHSs as opposed to including distal sites in the analysis in Fig. 2f? Especially since the authors mention the "importance of distal DHSs in defining cell identity".

Response to reviewers' comments

We would like to thank the reviewers for their constructive comments. We address their comments point-by-point below:

Referee #1 (Remarks to the Author):

In the manuscript, entitled "Gmeb1 is a transcriptional regulator important for dopamine neuron function," Tuesta, Zhang and colleagues present a series of compelling data—using sophisticated molecular tools and approaches—to demonstrate a potentially novel role for the Glucocorticoid modulatory element-binding protein 1 in the regulation of dopaminergic gene expression, and thus motor function, in brain. To do so, the authors first expressed an AAV5-DIO-KASH-HA virus into the midbrain (hitting both VTA and SNc) of DAT-cre mice, followed by HA associated FANS purifications, RNA-seq and low input DNase-seq to obtain gene expression and chromatin accessibility profiles specifically within DA neurons (vs. non-HA+ cells - mainly glial cells based on the seq data). In doing so, the authors provide very nice controls demonstrating the validity of this approach including comparisons of alternative neuronal subtypes in the cortex (e.g., VIP, Exc and PV neurons). Following identification of DAergic enriched gene expression and DHS, motif analyses were employed in an attempt to identify putative upstream TFs that may be important for mediating DAergic specific gene expression patterns; such assessments resulted in the identification of Gmeb1, alongside with other well known regulators of DAergic gene expression (e.g., Atf1 and Creb1), as a possible upstream regulator of DAergic genes, such as Th and Slc18a2. To next assess whether Gmeb1 itself is truly able to mediate these patterns of gene expression, conditional shRNA vectors targeting Gmeb1 were injected into the midbrain of DAT-Cre mice, followed again by FANS RNA-seq, as well as immunohistochemical validations of Dat expression in the dorsal striatum. Finally, to demonstrate that knocking down Gmeb1 in DAergic neurons, which reduces both Th and Dat expression, actually leads to functional consequences, the authors performed numerous assessments of motor outputs including Swim and Pole tests, Rotarod and the Hanging Wire test - in doing so, they validate that DAergic Gmeb1 KD does indeed lead to predicted deficits in striatal driven motor functions. Overall, this is a thorough, well designed and interesting manuscript that would be of broad interest to the field.

Response: We thank this reviewer for nicely summarizing our study and for the nice words on our work.

However, there are a few points that need to be addressed prior to publication in Nature Communications:

1) Perhaps I missed it, but is Gmeb1 preferentially enriched in DAergic neurons vs. the other neuronal subtypes assessed (or even vs. glia for that matter)? If so, then the overall claims of the manuscript seem reasonable; however, if this TF is more broadly expressed in brain (as would be suggested from the Allen Brain Atlas), then the authors should tone down their claims relating to the specificity of Gmeb1 activity in the mediation of DAergic gene expression, per se.

Response: We thank this reviewer for raising this important question, but we did not claim that Gmeb1 is preferentially enriched in mDA neurons. Indeed, based on RNA-seq data, Gmeb1 is broadly expressed in many cell types. Thus, its specific function in mDA neurons is unlikely to be controlled at the transcriptional level. However, given that previous studies have demonstrated that DNA methylation prevents Gmeb1 from binding to its targets (Burnett et al., 2001), we hypothesized that Gmeb1 binding motifs would be hypomethylated in mDA neurons, but hypermethylated in cortical neurons. To test this possibility, we performed whole genome bisulfite sequencing (WGBS) on purified mDA nuclei and compared it with available DNA methylation data sets of cortical neurons. Data shown below in **Fig. R1** indicate that the Gmeb1 putative binding site of *Th* is indeed hypomethylated in mDA neurons, but is hypermethylated in cortical neurons, supporting the notion that the specificity of Gmeb1 function is mediated through differential DNA methylation in different neuron types.

Fig. R1. Genome browser view of DNA methylation state of the *Th* promoter in mDA and cortical neurons. Note the hypomethylation state at the Gmeb1 binding site in mDA allows its binding to the *Th* promoter in mDA neurons.

2) While I greatly appreciate the numerous controls provided for the Gmeb1 KD studies, it would have been nice to also see a Gmeb1 overexpression experiment to demonstrate bidirectional effects of Gmeb1 manipulations on the expression of target genes (either by RNA-seq, or even with just a focused analysis of *Th* and *Dat*).

Response: Following this reviewer's suggestion, we overexpressed mouse Gmeb1 in the mouse Neuro-2A cell line and then performed RNA-seq with replicates. RT-qPCR analysis indicates that while Gmeb1 overexpression results in a ~15-fold increase of Gmeb1 mRNA, its effect on *Th* expression is modest (**R2**). These results indicate that Gmeb1 may not be limiting in the Neuro-2A cell line, thus the effect of Gmeb1 overexpression is modest. Alternatively, factors other than Gmeb1 might be required to activate *Th* expression.

Fig. R2. Overexpression of Gmeb1 in Neuro2A cells. Expression level (RPKM) of Gmeb1 (left panel) and Th (right panel). Overexpression vector (oeGmeb1); Control vector (ctrl).

3) Is Gmeb1's regulation of genes, such as Th and Dat, more of a homeostatic regulatory mechanism in adult brain, or do the authors envision that Gmeb1 is also responsible for activity-dependent regulation of these genes? In other words, it is surprising to me that KD of Gmeb1 alone leads to such dramatic decreases in Th and Dat expression, while other TFs, such as ATf1 and Creb1, are also thought to be involved in regulating their expression. Therefore, I am wondering if Gmeb1 may play more of a homeostatic maintenance function, while TFs like Creb1 may be more important for activity-dependent regulation. This could be tested by giving mice drugs (e.g., cocaine or amphetamine) or optogenetic/chemogenetic stimulations, that increase DA release from VTA into striatum—manipulations that also tend to lead to increased expression of Th and/or Dat in VTA—in the context Gmeb1 KD to see if stimulation induced gene expression of these genes is similarly controlled by Gmeb1 (or if these patterns of expression are more controlled by alternative TFs).

Response: This is an excellent question, which unfortunately cannot be fully addressed due to time constraints for resubmission, but would be an interesting question for a follow-up study. Nevertheless, our preliminary data suggest that the role of Gmeb1 on transcription regulation is likely to be homeostatic rather than activity-dependent. We found that the mRNA level of Gmeb1 in mDA neurons is not significantly altered following a 14-day course of cocaine administration (20 mg/kg, IP, daily) compared to control mice injected with saline, suggesting that *Gmeb1* expression is not regulated by chronic activation of mDA neurons (**R3**). Further, considering the modest increase in *Th* following *Gmeb1* overexpression (**R2**), we would hypothesize that any neuronal activity that would increase *Gmeb1* expression would not have as great a regulatory effect as loss of Gmeb1 on *Th* expression.

Fig. R3. No significant alteration of Gmeb1 after 14-day cocaine (20 mg/kg, IP) regimen.

4) Given the profound impact of Gmeb1 KD on *Dat* expression in mice (i.e., it looks to be almost completely ablated), I am curious whether the authors have performed measurements of either DA levels in striatum and/or firing activity of VTA DAergic neurons in the context of Gmeb1? With such reductions in gene expression and motor functions, are the author sure that the infected neurons remain healthy with Gmeb1 KD? This would be an important control for both downstream gene expression and behavioral studies.

Response: We thank this reviewer for this suggestion. However, our laboratory currently lacks the equipment to perform electrophysiological recordings or measure real-time DA release. The time required to prepare the mice and obtain the equipment through purchase and/or collaboration would greatly exceed the timeframe allowed for resubmission. However, we do recognize the importance of the health of mDA neurons in properly interpreting the behavioral outcomes reported in our study.

Indeed, we were also surprised to observe such a profound effect on *Th* and *Dat* expression in our Gmeb1 loss-of-function study (**Fig. 3**), which led us to consider that Gmeb1 loss could induce apoptosis in mDA neurons. However, RNA-seq analysis of Gmeb1 knockdown and control mDA neurons did not show any significant expression changes in apoptosis-related genes (**R4**). Furthermore, Gmeb1 knockdown does not significantly affect the level of cleaved caspase-3 in mDA neurons when compared to controls (**Supplementary Fig. 9a and b**), suggesting that loss of Gmeb1 does not induce cell death.

Fig. R4. Knockdown of Gmeb1 does not alter expression of apoptosis-related genes in mDA neurons. Note: These data can also be accessed in Supplementary figure 9a.

5) Finally, given the apparent critical importance of Gmeb1 expression in DAergic neurons in adulthood, I wonder if the authors have assessed its expression patterns or regulatory effects during DAergic neuronal development? For example, between E9 and E17 in mice, DAergic gene expression (including *Th* and *Dat*) are massively increased as DAergic neurons develop...is Gmeb1 important for these patterns of gene expression, or is it more involved in the maintenance of DAergic gene expression in adult animals?

Response: We thank the reviewer for this question. Indeed, *Nurr1* and *Foxa2*, two genes known to regulate transcription in mDA neurons are prominent developmental factors for this population (Hegarty et al., 2013), so it is reasonable to believe that a transcription factor that has a critical role in regulating *Th* and *Dat* expression should have commensurate effects on mDA neuron identity during embryonic development. However, since the nuclear tagging method requires viral infection of mDA neurons via stereotaxic injection, it would be technically impossible to perform such procedure on mice at that early stage of development. While we could generate conditional KO mice to study the developmental role of *Gmeb1*, such initiative would require an extended time commitment and thus would be more appropriate for a follow-up study. We want to point out that the major focus of our current study is to develop approaches and tools for isolating, and profiling pure adult neuron subtypes to understand how regulation of neuron subtype-specific gene expression is achieved. Identification of *Gmeb1* as a novel TF regulating mDA-specific genes *Th* and *Dat* demonstrates we have achieved our goal.

In sum, this is a generally exciting piece of work, however, the inclusion of additional studies (as discussed above) aimed at nailing down Gmeb1's role in DAergic gene expression would greatly strengthen the manuscript.

Reviewer #2 (Remarks to the Author):

The manuscript by Tuesta and colleagues combines transcriptome and chromatin accessibility analyses to identify *Gmeb1* as a new transcription factor regulating cell-specific gene expression in midbrain dopamine (mDA) neurons. Later, they use loss-of-function experiments (shRNA-based) to demonstrate that genes typically expressed in mDA cells, such as *Th* and *Dat*, are down-regulated, and that motor coordination results impaired.

The approach is technically innovative, interesting and could be applied to other discovery-driven projects in the nervous system. The follow-up proof-of-principle experiments are straightforward and support the notion that *Gmeb1* is important for mDA function, although they do not delineate the actual role of this transcription factor in the biology of mDA cells. My specific criticisms are below:

Response: We thank this review for nicely summarizing our study and for the kind words on our work.

1. The authors should check whether *Gmeb1* loss-of-function affects neuronal viability. They should present histological analysis and explore the possible death of mDA neurons. The images presented in Sup. Fig. S8 represent a very compelling evidence for the downregulation of DA-related genes. The authors may consider including some of these images in a main figure along with higher magnification images showing if the cells that do not longer express *Dat* and *Th* are dying or remain viable.

Response: We thank the reviewer for this comment. In our opinion, this was the main concern raised, as it was brought up by each reviewer. We would kindly direct the reviewer's attention to our response to reviewer #1, question #4.

With regard to this reviewer's specific suggestion, since similar results have been shown in Fig. 3, we choose not to make the change. With regard to the concern that *Gmeb1* knockdown causes cell death, our newly added **Supplementary figure 9** indicates this is not the case.

2. To further differentiate if *Gmeb1* is necessary for mDA cells viability or identity. The authors could consider including some electrophysiological experiments in neurons infected with AVV5-DIO-KASH-GFP-U6-sh*Gmeb1*. It would be very interesting to see whether these neurons have lost some mDA specific-features, but still behave as neurons (preserved resting potential, etc).

Response: We thank the reviewer for this suggestion, which is indeed an excellent idea for a follow-up study. Similar to our response to a previous comment on electrophysiology, our laboratory currently does not have the rig necessary to carry out these experiments and doing so, either through purchase or collaboration would well extend the resubmission window for this manuscript.

Minor issues:

3. Page 4: The description of GO enrichment analysis in the second paragraph should be revised. They indicate that differential profiling identified 107 genes; later they indicate that some genes were excluded from the list, and a few lines below they refer again to 107 genes. In which specific step and analyses were genes excluded? I should also note that the criterion for exclusion seems arbitrary and should be avoided in this type of analysis. GO analyses have some hypothesis-generating value, but should be conducted in an unbiased manner because the results of the analysis can be easily influenced by ad hoc filtering of the gene list. For example, the authors claim in page 4 that they “defined these 107 genes as ‘mDA-enriched genes’”. Did they reach that conclusion after manually excluding genes that were “highly expressed in cortical neuron subtypes? This is not clear in the text.

Response: It seems there is some misunderstanding here. All differential gene detection analyses were performed in an unbiased manner and we kept the same differential gene detection criteria throughout the manuscript (p -value < 0.001 and FC>4) unless otherwise specified. The GO analyses performed in our case were aimed more to assess whether the detected genes were enriched in mDA neuron-related functions rather than to gain new functional insights. Hence, our first GO analysis was performed to check whether the 394 genes enriched in HA+ nuclei were actually from mDA neurons – we needed to determine the cellular identity of our sample by transcriptional analysis. Next, to detect genes that were more specific to mDA neurons (mDA-enriched), we retained only those genes that showed at least a 4-fold enrichment compared to cortical neurons. Only 107 genes fit these criteria. Here we used 4-fold change, instead of the widely-used 2-fold change criteria, to be more stringent and selective for the most reliable genes.

4. Page 6: In the first paragraph the authors connect their genomic screen with the single gene study that occupies the second part of the manuscript. The reasons for focusing on *Gmeb1* are unclear. They refer to a “predictive scheme accounted for the motif p -value and distance from DHS to TSS” that is explained in greater detail in Methods. Given the importance of this selection step, the authors should explain better the basis of their “predictive scheme” to a non-expert audience. The description in methods is very cryptic.

Response: We apologize for the lack of clarity. We have edited this section to make it more understandable in the revised version. Briefly, we found that the *Gmeb1* motif along with the motif of other transcription factors was enriched in the accessible promoter (DHS) regions of 59 mDA-enriched genes. To determine which TF could regulate which gene (among the 59 mDA-enriched genes), we generated a predictive model based on genes that show a positive correlation between their expression level and 1) binding affinity of the TF, and 2) the proximity at which the TF binds from the TSS.

5. Page 8: The description of the authors suggests that swimming involves motor learning. However, swimming is an innate behavior in mice not a learned skill.

Response: The reviewer is correct in this assertion. However, we do not claim that the animal is learning a new skill, rather that the animal is adapting to a new environment

(inescapable water challenge) by employing an innate behavior that requires coordinated movement, and one that the animal has not performed. We have modified the sentence in the **results section** to avoid any confusion.

6. A Venn diagram does not seem the optimal representation for the results presented Figure 2a. As could only be expected, many intersection shows a 0 value. A sector graph seems more appropriate to present the percentage of genes unique for DA and those coincident with the different neuronal populations.

Response: We thank the reviewer for this excellent suggestion. Indeed, a pie chart would be the best way to present the data. **Fig. 2a** now reflects this change.

7. Introduction, 6 line: "...heterogeneity in (the) brain..."

Response: We thank the reviewer for the suggestion. The word has been inserted.

Reviewer #3 (Remarks to the Author):

In this study, the authors use a novel viral-based approach to label mDA nuclei, sort them, and perform gene expression and chromosome accessibility analysis. They use previously published data on three cortical cell populations to define mDA-enriched genes and DHSs. Through this approach and predictive modeling to define TF binding sites for mDA-enriched genes, the authors discover a novel TF, Gmeb1, that appears to regulate expression of dopaminergic genes including Th and Slc6a3. They further show that in vivo knock-down of Gmeb1 reduces Th protein levels in the midbrain and striatum and causes motor impairments. While this study has some interesting components, further work is needed to clarify some key points, as described below.

Response: We thank this reviewer for nicely summarizing our study and for the supportive comments on our work.

Major points:

-An important limitation of this study is the way that mDA-enriched genes and DHSs were defined. These were defined by comparison to a previously published data set on three unrelated populations of cortical neurons. It is likely that if the authors used different cells as filters for exclusion (i.e. more closely related serotonergic or midbrain GABA-ergic neurons) that they would have a different list of genes/DHSs. In addition, it is unclear that Gmeb1 is uniquely required for DA neuron function – it could be highly active in other cell types that were not profiled. Also, by defining enriched genes in this way, the authors exclude genes that are very important for DA neuron development/function (e.g. Nurr1) but that happen to be expressed in one of the three cortical cell types.

Response: We thank the reviewer for these comments. Indeed, we could have performed comparisons to other cell types that were more closely related to mDA neurons such as midbrain GABAergic or serotonergic neurons, as suggested. However, repeating the motif enrichment analysis on the promoter DHS of all the HA+enriched genes (394), yields similar conclusions (**R5**). Therefore, performing the same analysis with a much wider gene list still generated similar results to those generated by comparison to cortical neurons.

Target	PWM	P-value	In top motifs
Gmeb1		1.5e-18	35 %
Klf4		1.48e-14	23 %
Klf7		6.66e-10	22 %
Bhlhe41		3.79e-25	22 %
Atf1		1.73e-32	21 %
Sp4		5.49e-06	20 %
Creb1		2.51e-18	19 %
Atf1		7.35e-06	18 %
Bhlhe40		1.03e-09	18 %
GABPA		4.79e-05	18 %

Fig. R5. Motif analysis with 394-gene list. Gmeb1 still appears as the top enriched motif when comparing all promoter DHSs from the 394 HA+ enriched genes.

Also, while the reviewer is absolutely correct in this criticism, one limiting factor in comparing mDA neurons to serotonergic or midbrain GABAergic neurons is the lack of available open chromatin maps for those specific cell types with which to derive mDA-enriched DHSs. For our study, we used the available open chromatin maps and transcriptomes from cortical neurons generated by (Mo et al., 2015), mainly to prune away pan-neuronal genes and open chromatin regions, and to arrive at a list of mDA-enriched DHSs. To test the strength of our approach, we performed unbiased motif analyses (**Supplemental figure 4 b,c,d**) on open chromatin regions from those cortical neurons (Exc, PV, VIP) and found that *Gmeb1* did not appear as a candidate for any cortical cell type, suggesting that transcriptional regulation by *Gmeb1* may be more

specific to mDA neurons than to cortical cell types, which is supported by the data presented in **Fig. R1**.

-Related to the point above, it would be helpful if the authors could more clearly state their goal. Are they trying to define the TFs most important for DA neuron function? If so, then perhaps they should not filter genes based on expression in other cell types. Or, are the authors trying to identify TFs active in DA neurons exclusively? If this is the case, then a limited comparison to three cortical cell types does not rule out the possibility that the TFs they identify are active in other non-DA cells.

Response: The main goal of this study is by developing molecular tools and bioinformatics methodology to identify important dopaminergic transcriptional regulators. Identification of *Gmeb1* as a new regulator of *Th* and *Dat* expression and demonstration of its function indicate that we have achieved our goal.

-The identification of *Gmeb1* as a regulator of dopaminergic genes is interesting and novel. The authors perform proof-of-principle experiments that *Gmeb1* knockdown affects expression of genes such as *Th* and causes motor impairments; however more information is needed to support their conclusions (see specific comments below).

Specific points:

1) What is the endogenous expression pattern of *Gmeb1* in the midbrain? Is it expressed exclusively in DA neurons or also in surrounding astrocytes or non-DA neurons? Is it expressed equally in all DA neurons (i.e. VTA and SNc)? If it is expressed in non-DA cells, the observed phenotypes could be due to non-cell autonomous changes in other cell types that indirectly affect the health of the DA neurons. The authors could investigate *Gmeb1* expression patterns with immunohistochemistry (if there is a good antibody) or in situ hybridization and cell-type specific markers.

Response: A similar question was also raised by reviewer #1, please refer to our answer to question #1 of reviewer #1. A short answer is that *Gmeb1* is broadly expressed in many cell types (neuronal and non-neuronal cell types). Unfortunately, there are currently no good *Gmeb1* antibodies available. While we cannot rule out that the behavioral phenotypes may be due to non-cell autonomous changes in other cell types, the observed deficits in balance and coordination are highly consistent with deficits in dopamine signaling. Furthermore, we are confident that the transcriptional effects of *Gmeb1* knockdown on *Th* and *Dat* expression are cell autonomous, lending credence to our hypothesis that *Gmeb1* is a key regulator of dopamine neuron function and that its specificity is at least partly controlled by differential DNA methylation.

2) Are the DA neurons dying with *Gmeb1* knockdown? The authors could investigate this by crossing the DAT-Cre line to a tdTomato Cre reporter and/or examining markers of apoptosis at different time points post-injection.

Response: We thank the reviewer for this suggestion. With regard to the question of whether mDA neurons are dying with *Gmeb1* knockdown, we would like to refer this reviewer to our response to question #4 from reviewer #1. In short, RNA-seq analysis of mDA neurons with *Gmeb1* knockdown do not reveal significant changes in apoptosis-related genes (**R4**). Further, immunostaining shows no significant increase in cleaved caspase-3, an apoptosis marker (**Supplementary figure 9b, c**).

3) Since it's possible that the DA neurons are simply sick/dying, the authors can't formally conclude that "loss of dopamine caused by *Gmeb1* knockdown results in balance and coordination impairments" (pg 7) or "our results suggest that *Gmeb1* knockdown in the SNc leads to motor impairments due to the disruption of dopamine signaling" (pg 8). If the authors want to claim that the motor deficits are due to loss of dopamine synthesis specifically, they should measure dopamine levels directly with HPLC or another method.

Response: This is a good point. Indeed, while we cannot conclude that the behavioral effects are due to *Gmeb1* knockdown without first ruling out neuronal injury/death, our transcriptome (**R4**) and histological analysis (**Supplementary figure 9b, c**) suggest that *Gmeb1* knockdown does not induce apoptosis in mDA neurons. Considering that Th is the rate-limiting step in DA synthesis, it is very likely that a near abrogation of the enzyme would cause a disruption in the production of the neurotransmitter, and as such result in motor deficits. However, we understand that our original statement could be too far-reaching without the appropriate measurements (DA release in striatum). As such, we have amended the manuscript to reflect a more conservative conclusion.

4) In Fig 3d, the authors only show an image of the VTA, but claim that the motor impairments are driven by changes in SNc DA neurons. In addition, it appears that the KASH-GFP is being expressed in non-DA cells (Th-negative) – if so, then this can't be used as a way to show that the DA neurons have not degenerated.

Response: We thank the reviewer for raising this point. The main reason that we imaged the VTA to show Cre-specificity (**Fig. 3d**) is because it flanks the IPN on both sides (outlined by white dotted line). The IPN does not contain mDA neurons. Therefore, it should not (and does not) express KASH-GFP. Considering that the virus will always spread into the IPN, a negative signal from this brain region will serve as an anatomical control to ensure Cre-specificity for DA. With regard to the observation that some KASH-GFP neurons do not express Th. This is likely due to 1) low expression of Th in those neurons, and 2) low image resolution in the original submission. To this end, we have replaced **Fig. 3d** with a higher-resolution panel.

5) Are the *Gmeb1* knock-down mice hypoactive? Multiple motor parameters including locomotor activity could be measured with a simple open field test.

Response: This is a great suggestion. Thus we measured the locomotor activity of *Gmeb1* knockdown and control mice in an open-field locomotion arena during a 30-minute session. Data presented in **R6** indicate that knocking down *Gmeb1* (SNc) did not

produce a hypoactive phenotype (n=5 per group) as indicated by the distance traveled over the 30 min (cm). These data can also be accessed in **Supplementary figure 10**.

Fig. R6. Open field locomotion. Mice with bilateral SNc knockdown of *Gmeb1* (or scrambled control) were allowed to explore the insides of an open field locomotion testing apparatus for 30 minutes, during which, their movement was recorded by beam breaks as distance traveled (cm).

6) To show that the motor impairments and gene expression changes are specific to *Gmeb1* (and downregulation of genes controlling DA synthesis), the authors could perform injections of shRNA against another transcription factor, which is not expected to regulate Th expression (e.g. *Tcf3* or *Clock*). This would help confirm the predictions made in Fig. 2f.

Response: As a control for the profiling and behavioral experiments we used a scrambled shRNA. However, targeting alternate TFs not predicted to regulate Th expression would serve as very nice additional controls. Unfortunately targeting *Tcf3* or *Clock* would require a time commitment beyond the time allotment for resubmission of the revised manuscript. We hope the reviewer can understand our decision to not pursue this suggestion.

7) The authors should cite a relevant dopamine neuron single cell RNA-sequencing study in the discussion (pg 8) – La Manno et al, Cell, 2016. Reference #10 (Poulin et al, 2014) did not perform single-cell RNA-seq but rather used single-cell qPCR for 96 candidate genes.

Response: We thank this reviewer for catching this oversight. We have made the correction on the revised manuscript.

8) What is the rationale for only focusing on promoter DHSs as opposed to including distal sites in the analysis in Fig. 2f? Especially since the authors mention the “importance of distal DHSs in defining cell identity”.

Response: Gene expression is regulated through the interplay between promoters and distal regulatory regions. Hence, we initially thought that the independent analyses of promoters and distal DHSs would yield a more nuanced view of the TF cis- and trans-

regulatory networks involved in the regulation of mDA neuron gene expression. While distal DHS motif analysis revealed that *Nurr1* and *Foxa2* operate mainly through distal regulation (which to our knowledge was not reported before), their role in mDA function has already been established. On the other hand, promoter DHS analyses revealed the enrichment of some factors previously unknown to play a role in mDA transcriptional regulation, such as *Gmeb1*. Characterizing these factors would have more novelty.

We thank the reviewers once again for their insightful comments, which have helped us greatly improve our manuscript. We hope that the reviewers find our responses satisfying.

Sincerely,

Yi Zhang, Ph.D.

References

- Burnett, E., Christensen, J., and Tattersall, P. (2001). A consensus DNA recognition motif for two KDWK transcription factors identifies flexible-length, CpG-methylation sensitive cognate binding sites in the majority of human promoters. *J Mol Biol* 314, 1029-1039.
- Hegarty, S.V., Sullivan, A.M., and O'Keeffe, G.W. (2013). Midbrain dopaminergic neurons: a review of the molecular circuitry that regulates their development. *Developmental biology* 379, 123-138.
- Mo, A., Mukamel, E.A., Davis, F.P., Luo, C., Henry, G.L., Picard, S., Urich, M.A., Nery, J.R., Sejnowski, T.J., Lister, R., *et al.* (2015). Epigenomic Signatures of Neuronal Diversity in the Mammalian Brain. *Neuron* 86, 1369-1384.

Reviewers' comments:

Reviewer #1 (Remarks to the Author):

The authors have done a nice job at responding to my previous critiques, as well as those from the other Reviewers. While I continue to feel that it would have been nice to try and tackle the issue of whether Gmeb1 is associated more with homeostatic vs. activity-dependent regulation of DAergic gene expression, I agree with the authors' assessment that this may be beyond the scope of the current manuscript. In any case, I feel that the manuscript is now suitable for publication in Nature Communications.

Reviewer #2 (Remarks to the Author):

The revised manuscript includes a few minor changes to improve readability and a new supplementary figure addressing a common concern expressed by the three reviewers. In their rebuttal, the authors elude some of the most relevant criticisms referring to the limited timeframe allowed for resubmission. I found particularly disappointing that they did not try to evaluate the electrophysiological phenotype of cells with Gmeb1 KD (which also show green fluorescence). Such evaluation, requested by Reviewer#1 and myself, would significantly strengthen the conclusions of the study.

Regarding the only addition to the revised manuscript (Sup. Fig. 9), the authors only examined possible transcriptional changes in a short list of genes related to apoptosis (some of which are not even expected to be changed at the transcript level in dying cells). Neurodegeneration and neuronal death does not always occur through apoptosis. Since they have transcriptome information, they could also explore inflammation and alternative death pathways. Also, the cleaved caspase-3 staining presented in panel C is lacking a positive control. The detection of activation of caspase-3 is challenging *in vivo*. TUNEL staining and evaluation of gliosis in the affected tissue could be used to assess other pathological signs commonly detected in diseased brain tissue. A high magnification image of the nuclei of green neurons compared to neighboring neurons could also reveal differences in the chromatin (DAPI counterstain) associated with neuronal death.

Reviewer #3 (Remarks to the Author):

In this study the authors develop a novel viral approach that enables isolation of nuclei from specific cell types in the mouse brain. They apply this approach to dopamine neurons and perform gene expression analysis and accessible chromatin site mapping. They also develop a bioinformatics framework to predict transcription factors that may be specifically important for the function of dopamine neurons, as compared to three cortical neurons types. This aspect of the study appears to be well done (although this is not my area of expertise). Although the approach of nuclei sorting and isolation from defined cell types in the mouse brain is not novel, the viral-based DIO-KASH-HA construct is, to my knowledge. The authors use this workflow to identify a transcription factor, Gmeb1, which they predict to regulate key dopaminergic genes including TH and DAT. Gmeb1 has not previously been linked with dopamine neuron function. To test whether maintained expression of Gmeb1 is required for DA neuron function, they use shRNA to knock it down *in vivo* and examine several motor behaviors that are sensitive to dopamine function. They find deficits in climbing, swimming, and balance/coordination with no change in muscle strength or general locomotion. This aspect of the study, while potentially interesting, still feels quite preliminary. In particular, there are a few key points that still need to be addressed.

First is the issue of whether the DA neurons are dying. The authors report levels of apoptotic genes and find no difference with Gmeb1 knock-down, however, this (and caspase-3 staining) may not be a sensitive way to determine this, especially if the neurons have already degenerated (in 6-OHDA models, DA neurons can be mostly dead by 2 weeks). It would be much better if the authors could perform a simple cell count (with a few mice as replicates) of DA neurons (SNc + VTA) following injection of scrambled versus Gmeb1 shRNA. They could use the tdTomato signal in the DAT-Cre;Ai9 mice, which would not be confounded by low TH/DAT levels. It would also be helpful to analyze tdTomato+ projections in the striatum as axonal denervation is one of the first steps in degeneration.

Since they are knocking down a transcription factor that regulates many genes, major effects on cell health might be expected. The authors chose not to investigate the consequences of knocking down another TF in these cells, which I understand would be a time-consuming endeavor. However, it is possible that knocking down any major TF in DA neurons could have a similar detrimental effect on cell health/function. This possibility should be raised as a caveat when interpreting the authors' findings.

Since the only behavior phenotypes measured in this study are motor, the authors must show an image(s) of the SNc in Gmeb1 shRNA mice compared to control. Their argument for showing only the VTA is not convincing as the SNc is also surrounded by non-dopaminergic cells. It is important to show that the dopamine neurons in the SNc are still alive and expressing the KASH-HA construct. Even with the improved resolution VTA images, it still appears that non-TH+ cells express this construct which is concerning (this is observable with scrambled construct, which should not have low TH levels as a confound).

The authors have run the open field test to measure locomotor activity as suggested, however, with an n of 5, they are underpowered to find statistically significant effects. Behavior experiments are highly variable and usually an n of 12+ mice is required for statistical confidence. It appears that the Gmeb1 shRNA mice may indeed be hypolocomotor but with this low n it's difficult to conclude either way. Note that the other behavior tests do not measure locomotor activity per se, rather they assess different aspects of motor behavior/balance/coordination.

Minor points:

- 1) In line 208, the authors imply that loss of dopamine is associated with drug addiction, however, drugs of abuse act to increase dopamine signaling.
- 2) The claim in lines 255-256 that the behavior phenotypes are in part due to downregulation of TH is overstated as these behavior changes could result from a number of different mechanisms, both cell autonomous and non-autonomous. The authors would need further experiments to claim that these phenotypes are due to downregulation of TH specifically.
- 3) The analysis presented in Fig R5 is important and shows that their results are not overly biased by comparison to the three cortical neuron types. It would be helpful if the authors could include this analysis in the supplementary information.
- 4) The authors should state in the text that Gmeb1 is expressed in non-DA neurons as well.

Response to reviewers' comments

We would like to thank the reviewers for their constructive comments. We address their comments point-by-point below:

Reviewer #1 (Remarks to the Author):

The authors have done a nice job at responding to my previous critiques, as well as those from the other Reviewers. While I continue to feel that it would have been nice to try and tackle the issue of whether *Gmeb1* is associated more with homeostatic vs. activity-dependent regulation of DAergic gene expression, I agree with the authors assessment that this may be beyond the scope of the current manuscript. In any case, I feel that the manuscript is now suitable for publication in Nature Communications.

Response: We thank this reviewer for his/her valuable feedback and are appreciative of his/her view that our manuscript is now suitable for publication in Nature Communications.

Reviewer #2 (Remarks to the Author):

The revised manuscript includes a few minor changes to improve readability and a new supplementary figure addressing a common concern expressed by the three reviewers. In their rebuttal, the authors elude some of the most relevant criticisms referring to the limited timeframe allowed for resubmission. I found particularly disappointing that they did not try to evaluate the electrophysiological phenotype of cells with *Gmeb1* KD (which also show green fluorescence). Such evaluation, requested by Reviewer#1 and myself, would significantly strengthen the conclusions of the study.

Response: We understand that this reviewer is concerned whether *Gmeb1* KD affects the basic electrophysiology of mDA neurons. To this end, we collaborated with the Sabatini lab at HMS and found that *Gmeb1* knockdown SNc mDA neurons indeed did not affect baseline electrophysiological properties such as resting potential, AP threshold voltage, time to AP threshold, and peak AP voltage (**Supplementary Fig. 11a**). Further, *Gmeb1* knockdown also did not affect the frequency, nor amplitude of evoked APs (**Fig. 4**). These new results suggest that while *Gmeb1* reduces *Th* and *Dat* expression, the basic electrophysiology of SNc mDA is unaffected.

Regarding the only addition to the revised manuscript (Sup. Fig. 9), the authors only examined possible transcriptional changes in a short list of genes related to apoptosis (some of which are not even expected to be changed at the transcript level in dying cells). Neurodegeneration and neuronal death does not always occur through apoptosis. Since they have transcriptome information, they could also explore inflammation and alternative death pathways. Also, the cleaved caspase-3 staining presented in panel C is lacking a positive control. The detection of activation of caspase-3 is challenging in vivo. TUNEL staining and evaluation of gliosis in the affected tissue could be used to assess other pathological signs commonly detected in diseased brain tissue. A high magnification image of the nuclei of green neurons compared to neighboring

neurons could also reveal differences in the chromatin (DAPI counterstain) associated with neuronal death.

Response: The reviewer raises a valid point that neurodegeneration does not always occur through apoptosis and that cell death can occur through inflammatory mechanisms. To this end, we assessed the effects of *Gmeb1* knockdown in mDA neurons on inflammatory gene expression. **Supplementary Fig. 9d** shows that inflammatory gene expression remains unaffected in *Gmeb1* knockdown mDA neurons when compared to shScramble controls. Further, we also performed the suggested TUNEL assay on midbrain slices of Dat-Cre mice co-injected with a 1:1 mix of AAV-DIO-mCherry with either shGmeb1 or shScramble virus (SNc). We found similar numbers of mCherry+ cells (mDA neurons) in both injection groups, suggesting that while *Gmeb1* knockdown reduces Th expression, no clear sign of increased mDA neurons death (**Supplementary Fig. 10c**). Further, TUNEL assay showed low levels of fragmented DNA in both groups, while DNase-1 treatment (positive control) resulted in robust TUNEL signal (**Supplementary Fig. 10a**). Collectively, our results suggest that *Gmeb1* knockdown does not affect mDA neuron viability, as evidenced by unchanged expression of apoptosis- and inflammation-related genes, lack of fragmented DNA, comparable mDA neuron cell counts between *Gmeb1* knockdown and shScramble controls, and more robustly, by our electrophysiology data.

Reviewer #3 (Remarks to the Author):

In this study the authors develop a novel viral approach that enables isolation of nuclei from specific cell types in the mouse brain. They apply this approach to dopamine neurons and perform gene expression analysis and accessible chromatin site mapping. They also develop a bioinformatics framework to predict transcription factors that may be specifically important for the function of dopamine neurons, as compared to three cortical neurons types. This aspect of the study appears to be well done (although this is not my area of expertise). Although the approach of nuclei sorting and isolation from defined cell types in the mouse brain is not novel, the viral-based DIO-KASH-HA construct is, to my knowledge. The authors use this workflow to identify a transcription factor, *Gmeb1*, which they predict to regulate key dopaminergic genes including TH and DAT. *Gmeb1* has not previously been linked with dopamine neuron function. To test whether maintained expression of *Gmeb1* is required for DA neuron function, they use shRNA to knock it down in vivo and examine several motor behaviors that are sensitive to dopamine function. They find deficits in climbing, swimming, and balance/coordination with no change in muscle strength or general locomotion. This aspect of the study, while potentially interesting, still feels quite preliminary. In particular, there are a few key points that still need to be addressed.

First is the issue of whether the DA neurons are dying. The authors report levels of apoptotic genes and find no difference with *Gmeb1* knock-down, however, this (and caspase-3 staining) may not be a sensitive way to determine this, especially if the neurons have already degenerated (in 6-OHDA models, DA neurons can be mostly dead by 2 weeks). It would be much better if the authors could perform a simple cell count (with a few mice as replicates) of DA neurons (SNc + VTA) following injection of scrambled versus *Gmeb1* shRNA. They could use the tdTomato

signal in the DAT-Cre;Ai9 mice, which would not be confounded by low TH/DAT levels. It would also be helpful to analyze tdTomato+ projections in the striatum as axonal denervation is one of the first steps in degeneration.

Response: We thank this reviewer for this suggestion. Following this reviewer's suggestion, we co-injected a 1:1 mix of AAV-DIO-mCherry with either the shGmeb1 or shScramble virus in the SNc of Dat-Cre mice. Cell counting of the SNc sections using 3 mice per group (6 sections per mouse) revealed no significant difference in the number of mDA neurons following *Gmeb1* knockdown compared to that of control shScramble injection (**Supplementary Fig. 10c**). As expected, we also detected loss of Th signal in the *Gmeb1* knockdown group, indicating loss of Gmeb1, while reducing Th expression, does not alter the number of mDA neurons. Collectively, our results suggest that *Gmeb1* knockdown does not affect mDA neuron viability, as evidenced by unchanged expression of apoptosis- and inflammation-related genes, lack of fragmented DNA, and similar mDA neuron cell counts between *Gmeb1* knockdown and shScramble controls.

Since they are knocking down a transcription factor that regulates many genes, major effects on cell health might be expected. The authors chose not to investigate the consequences of knocking down another TF in these cells, which I understand would be a time-consuming endeavor. However, it is possible that knocking down any major TF in DA neurons could have a similar detrimental effect on cell health/function. This possibility should be raised as a caveat when interpreting the authors' findings.

Response: We appreciate this reviewer's concern. However, such a concern does not apply to the Gmeb1 knockdown as data presented in **Fig. 3b** clearly showed that Gmeb1 knockdown only affected the expression of very limited number of genes. This further supports our conclusion that Gmeb1 knockdown does not cause a significant increase in cell death, which can cause a global transcriptome change.

Since the only behavior phenotypes measured in this study are motor, the authors must show an image(s) of the SNc in *Gmeb1* shRNA mice compared to control. Their argument for showing only the VTA is not convincing as the SNc is also surrounded by non-dopaminergic cells. It is important to show that the dopamine neurons in the SNc are still alive and expressing the KASH-HA construct. Even with the improved resolution VTA images, it still appears that non-TH+ cells express this construct which is concerning (this is observable with scrambled construct, which should not have low TH levels as a confound).

Response: We thank this reviewer for the suggestion and we have performed the requested experiment. Data presented in **Supplementary Fig. 10a** indicate that knockdown of *Gmeb1* in SNc, while affecting Th expression, did not affect mDA neuron viability as evidenced by the presence of mCherry+ neurons.

The authors have run the open field test to measure locomotor activity as suggested, however, with an n of 5, they are underpowered to find statistically significant effects. Behavior experiments are highly variable and usually an n of 12+ mice is required for statistical confidence. It appears that the *Gmeb1* shRNA mice may indeed be hypolocomotor but with this

low n it's difficult to conclude either way. Note that the other behavior tests do not measure locomotor activity per se, rather they assess different aspects of motor behavior/balance/coordination.

Response: We thank the reviewer for raising this point. To address this concern, we added 4 mice to each group (shGmeb1, shScramble) and show that there is still no significant difference in locomotor activity between the two groups, as evidenced by distance traveled (cm) (n=9 for each group, **Fig. R1, Supplementary Fig. 12**, with the new cohort of 4 mice shown in green. As to the observation that the other tests do not measure “locomotor activity per se”, this reviewer is correct. We have amended the manuscript to highlight this distinction.

Figure R1: Open-field locomotion. Mean (\pm s.e.m.) distance traveled by mice in an open-field arena. Mice were allowed to explore the entirety of an open field arena for 30 minutes during which time the distance traveled (cm) was recorded as beam breaks, and the individual scores were averaged per group (n=9, $p=0.2639$ in student's t-test).

Note: Data points in green represent new cohort referenced in the response to this reviewer's comment.

Further, we also tested these mice in the locomotor battery we reported in previous versions of this manuscript (pole test, rotarod, swim test, hanging wire test) (n=11 per group). Behavioral scores in these tests were consistent with previous results and lowered the p -score in all instances (**Fig. R2, Fig. 5**). For detail, we include the new **Fig. 5** below with the new data points shown in green.

Figure R2: Midbrain *Gmeb1* knockdown results in locomotor impairments (a) Dot plot representing mean (\pm s.e.m.) time (s) required for mice with midbrain knockdown of *Gmeb1* or shScramble control mice to reach bottom of pole (n=11 per group, **** $p<0.0001$ in unpaired T-test). (b) Dot plot representing mean (\pm s.e.m.) latency (s) at which mice with midbrain knockdown of *Gmeb1* or shScramble control mice fall from rotarod or cling to apparatus without active locomotion (n=11 per group, **** $p<0.0001$ in unpaired T-test). (c) Dot plot representing mean (\pm s.e.m.) score reflecting the ability of *Gmeb1* or shScramble to swim over a 5-minute period (n=11 per group, **** $p<0.0001$ in unpaired T-test). (d) Dot plot representing mean (\pm s.e.m.) latency (s) at which mice with midbrain knockdown of *Gmeb1* or shScramble control mice fall from horizontal wire (n=11 per group, $p=0.3462$ in unpaired T-test).

Note: Data points in green represent new cohort referenced in the response to this reviewer's comment.

Minor points:

1) In line 208, the authors imply that loss of dopamine is associated with drug addiction,

however, drugs of abuse act to increase dopamine signaling.

Response: Indeed, drugs of abuse increase dopamine signaling, and chronic use dysregulates homeostatic dopamine signaling. We regret for the confusion and have modified this section of the manuscript for clarification.

2) The claim in lines 255-256 that the behavior phenotypes are in part due to downregulation of TH is overstated as these behavior changes could result from a number of different mechanisms, both cell autonomous and non-autonomous. The authors would need further experiments to claim that these phenotypes are due to downregulation of TH specifically.

Response: We thank the reviewer for this comment. Given the loss of Th following *Gmeb1* knockdown, and the referenced studies that use models of PD reported similar behavioral phenotypes, it is reasonable to conclude that our behavioral phenotype is at least partly due to loss of Th. However, the reviewer is correct to point out that we cannot rule out the possibility that the behavioral phenotype is in part due to non-DA effects. We have modified the manuscript to include this caveat.

3) The analysis presented in Fig R5 is important and shows that their results are not overly biased by comparison to the three cortical neuron types. It would be helpful if the authors could include this analysis in the supplementary information.

Response: We thank the reviewer for the suggestion and have included the data in the manuscript as **Supplementary Fig. 4b**

4) The authors should state in the text that *Gmeb1* is expressed in non-DA neurons as well.

Response: We thank the reviewer for raising this important issue. We have now included this fact in the **Results** section, under the heading “**Multi-omics analysis predicts *Gmeb1* as an mDA transcriptional regulator.**”

We thank the reviewers once again for their insightful comments, which have helped us improve our manuscript greatly. We hope that the reviewers find our responses satisfying.

Sincerely,

Yi Zhang, Ph.D

REVIEWERS' COMMENTS:

Reviewer #2 (Remarks to the Author):

The authors addressed all my concern in the second revision.

Reviewer #3 (Remarks to the Author):

In the revised version of the manuscript, the authors have addressed the major concerns related to the health of the Gmeb1 knock-down cells with additional assays, analyses and electrophysiology experiments. These new data strongly support the claim that the cells are likely not dying but rather have selectively reduced Th and DAT expression. The authors have also added more animals to the behavior analyses, which bolster their conclusions. In my opinion, the revised manuscript is ready for publication if a few minor issues are addressed:

1) I am still concerned about Figure 3d, in that there are a lot of KASH+ cells that are not expressing TH (it looks like 20-30% of KASH+ cells are TH-) - this contrasts with the quantification in Fig. 1, which shows high specificity of expression (although this is a different construct). If this image is representative of the expression pattern of this construct, the authors should state the caveat that they have reduced Gmeb1 expression in non-DA cells as well. While this is not expected to be a major confound to their results, it should still be discussed.

2) The sentence beginning on pg 313 seems rather speculative "This suggests that while Gmeb1 knockdown may not affect the ability of mDA neurons to communicate, due to the essential role of Th in dopamine synthesis, their reduced Th levels may render them "mute" and thus compromise their basic ability to function as dopamine neurons." Dopamine synthesis and release can be decoupled, therefore conclusions about the ability of these cells to "communicate" and whether or not they are "mute" cannot really be made with the current data (i.e. there may be increases or decreases pre-synaptic release that are independent of action potential properties and DA synthesis).

3) In the results the authors write on pg 8 "To this end, we tested mice with bilateral SNc Gmeb1 knockdown in a battery of locomotor assessments, including the pole test, rotarod test, swim test and hanging wire test." - further they discuss locomotion in the context of the swimming deficits. Perhaps just "motor assessments" or "motor deficits" would be better terminology to use.

4) In Fig. 4, what does "(cell 5-8)" and "(cell 5-11)" refer to? Perhaps this should be taken out. It would be helpful if the authors could show a representative single action potential trace from each group.

Response to reviewers' comments

We would like to thank the reviewers once more for taking the time to evaluate our work and provide constructive feedback. We address their comments point-by-point below:

Reviewer #2 (Remarks to the Author):

The authors addressed all my concern in the second revision.

Response: We thank this reviewer for his/her valuable feedback and are appreciative of his/her view that our manuscript is now suitable for publication in Nature Communications.

Reviewer #3 (Remarks to the Author):

In the revised version of the manuscript, the authors have addressed the major concerns related to the health of the Gmeb1 knock-down cells with additional assays, analyses and electrophysiology experiments. These new data strongly support the claim that the cells are likely not dying but rather have selectively reduced Th and DAT expression. The authors have also added more animals to the behavior analyses, which bolster their conclusions. In my opinion, the revised manuscript is ready for publication if a few minor issues are addressed:

1) I am still concerned about Figure 3d, in that there are a lot of KASH+ cells that are not expressing TH (it looks like 20-30% of KASH+ cells are TH-) - this contrasts with the quantification in Fig. 1, which shows high specificity of expression (although this is a different construct). If this image is representative of the expression pattern of this construct, the authors should state the caveat that they have reduced Gmeb1 expression in non-DA cells as well. While this is not expected to be a major confound to their results, it should still be discussed.

Response: We thank the reviewer for raising this issue and we agree that it should not go unaddressed. Indeed, the vector described in **Fig. 1** is different from the one used in **Fig. 3** – they use different backbones. The knockdown vector in **Fig. 3**, while encoding for KASH, uses GFP as a reporter and is driven by a human synapsin (hSyn) promoter whereas the vector in **Fig. 1** uses an HA reporter and is driven by an EF1 α promoter. These differences, particularly in the promoter, could affect the infection rate and while one could redesign the vector to more closely mirror the tagging vector used in **Fig. 1**, as the reviewer suggests, it would not affect the conclusions in the study. Regardless, we agree with the reviewer and have provided a brief explanation in the **Results** section of the manuscript.

2) The sentence beginning on pg 313 seems rather speculative "This suggests that while Gmeb1 knockdown may not affect the ability of mDA neurons to communicate, due to the essential role of Th in dopamine synthesis, their reduced Th levels may render them "mute" and thus compromise their basic ability to function as dopamine neurons." Dopamine synthesis and release can be decoupled, therefore conclusions about the ability of these cells to "communicate" and whether or not they are "mute" cannot really be made with the current data (i.e. there may be increases or decreases pre-synaptic release that are independent of action potential properties and DA synthesis).

Response: The reviewer is correct in making this distinction. We have rewritten this sentence to rein in this conclusion and state that the lack of tyrosine hydroxylase might compromise homeostatic dopamine signaling, which is evidenced by the behavioral data following Gmeb1 knockdown.

3) In the results the authors write on pg 8 "To this end, we tested mice with bilateral SNc Gmeb1 knockdown in a battery of locomotor assessments, including the pole test, rotarod test, swim test and hanging wire test." - further they discuss locomotion in the context of the swimming deficits. Perhaps just "motor assessments" or "motor deficits" would be better terminology to use.

Response: We agree with the reviewer on this point and have made the appropriate changes to the manuscript. More specifically, we now use "motor" in place of "locomotor" where appropriate.

4) In Fig. 4, what does "(cell 5-8)" and "(cell 5-11)" refer to? Perhaps this should be taken out. It would be helpful if the authors could show a representative single action potential trace from each group.

Response: The labels (i.e. cell 5-8, 5-11) used in **Fig. 4** refer to the IDs of the individual cells recorded. While this is useful for record-keeping, we agree that it could be a bit confusing to the reader. To simplify this figure, we have removed these designations and as suggested by the Reviewer, we are now showing a single representative trace from each group.

We thank the reviewers once more for their comments in helping us refine the quality of our manuscript.

Sincerely,

Yi Zhang, Ph.D